

# Measurement of ambient $NO_3$ reactivity: Design, characterization and first deployment of a new instrument

Jonathan M. Liebmann[1], Gerhard Schuster[1], Jan B. Schuladen[1], Nicolas Sobanski[1], Jos Lelieveld[1] and John N. Crowley[1]

[1]Atmospheric Chemistry Department, Max-Planck-Institut für Chemie, 55128 Mainz, Germany.

*Correspondence to*: John N. Crowley (john.crowley@mpic.de)

**Abstract.** We describe the first instrument for measurement of the rate constant ($s^{-1}$) for reactive loss (i.e. the total reactivity) of $NO_3$ in ambient air. Cavity-ring-down spectroscopy is used to monitor the mixing ratio of synthetically generated $NO_3$ ($\approx$ 30-50 pptv) after passing through a flow-tube reactor with variable residence time (generally 10.5 s). The change in concentration of $NO_3$ upon modulation of the bath gas between zero-air and ambient air is used to derive its loss rate constant, which is then corrected for formation and decomposition of $N_2O_5$ via numerical simulation. The instrument is calibrated and characterized using known amounts of NO and $NO_2$ and tested in the laboratory with an isoprene standard. The lowest reactivity that can be detected (defined by the stability of the $NO_3$ source, instrumental parameters and $NO_2$ mixing ratios) is 0.005 $s^{-1}$. An automated dilution procedure enables measurement of $NO_3$ reactivities up to 45 $s^{-1}$, this upper limit being defined mainly by the dilution accuracy. The typical total uncertainty associated with the reactivity measurement at the centre of its dynamic range is 16 %, though this is dependent on ambient $NO_2$ levels. Results from the first successful deployment of the instrument at a forested mountain site with urban influence are shown and future developments outlined.




## 1 Introduction

Large amounts of biogenic and anthropogenic trace gases are emitted annually into the atmosphere. Recent estimates (Guenther et al., 2012) suggest that about 1000 Tg of biogenic volatile organic compounds (bVOC), especially isoprene (contributing 50 %) and monoterpenes (15 %) are emitted annually by vegetation. The global burden of anthropogenic emission is dominated by $CO_2$, CO, $N_2O$, $CH_4$, $SO_2$, $NO_2$ and organic carbon, the latter contributing about 11 Tg (Huang et al., 2015). In particular, nitrogen oxides from combustion and microbial activity in soils have a major impact on the chemistry of the natural atmosphere (Crutzen, 1973). Most VOCs are oxidized efficiently in the Earth's boundary layer, the oxidizing capacity of which represents 15% of that of the entire atmosphere (Lelieveld et al., 2016). Biogenic and anthropogenic VOCs have a significant impact on air quality and human health and knowing and understanding their lifetimes, which are determined by the oxidizing capacity of the atmosphere, is prerequisite to predicting future atmospheric composition and related climate phenomena (Lelieveld et al., 2008).

During day-time, photo-chemically formed OH radicals represent the dominant contribution to the oxidative capacity of the atmosphere. As OH levels are vastly reduced in the absence of sunlight, the $NO_3$ radical (formed by reaction of $NO_2$ with $O_3$, R1) is the major oxidizing agent for many biogenic terpenoids and other unsaturated compounds at night-time (Brown and Stutz, 2012; Ng et al., 2016; Wayne et al., 1991; Atkinson, 2000; Atkinson and Arey, 2003a, b).

$$NO_2 + O_3 \rightarrow NO_3 + O_2 \tag{R1}$$

$NO_3$ reacts rapidly with NO (R2, rate constant 2.6 x $10^{-11}$ $cm^3$ molecule$^{-1}$ $s^{-2}$ at 298 K (Atkinson et al., 2004)) and undergoes rapid photolysis (R5, R6) so that its lifetime is usually of the order of seconds during the day and its concentration too low for it to be considered an important day-time oxidant.

At night, $NO_3$ can react with $NO_2$ forming $N_2O_5$, which thermally decomposes to set up a thermal equilibrium between $NO_2$, $NO_3$ and $N_2O_5$ (R3, R4) with $N_2O_5$ formation favoured by lower temperatures. As both $NO_3$ and $N_2O_5$ are formed from $NO_x$ ($NO_x$ = NO + $NO_2$) the loss of either $NO_3$ via gas-phase losses or $N_2O_5$ via heterogeneous uptake to particles or deposition implies a reduction in $NO_x$, and thus a reduction in the rate of photochemical $O_3$ formation (Dentener and Crutzen, 1993). In addition, heterogeneous loss of $N_2O_5$ can also result in release of $ClNO_2$ from chloride containing particles (R7) (Phillips et al., 2012; Mielke et al., 2011; Osthoff et al., 2008; Riedel et al., 2012; Thornton et al., 2010). The main loss processes of $NO_3$ are summarised in Fig. 1.

$$NO_3 + NO \rightarrow 2\,NO_2 \tag{R2}$$

$$NO_2 + NO_3 + M \rightarrow N_2O_5 + M \tag{R3}$$

$$N_2O_5 + M \rightarrow NO_3 + NO_2 + M \tag{R4}$$

$$NO_3 + h\nu \rightarrow NO_2 + O \tag{R5}$$

$$NO_3 + h\nu \rightarrow NO + O_2 \tag{R6}$$

$$N_2O_5 + surface \rightarrow NO_3^- \ (\text{and/or}\ ClNO_2) \tag{R7}$$





In rural and forested areas reaction with biogenic VOCs can dominate the loss of $NO_3$ (Mogensen et al., 2015). Especially
terpenoids like limonene ($k = 1.2 \times 10^{-11}$ $cm^3$ molecule$^{-1}$ s$^{-1}$), α-pinene ($k = 6.2 \times 10^{-12}$ $cm^3$ molecule$^{-1}$ s$^{-1}$) and isoprene ($k =$
$6.5 \times 10^{-13}$ $cm^3$ molecule$^{-1}$ s$^{-1}$) have high rate constants for reaction with $NO_3$ (IUPAC, 2016; Ng et al., 2016). Under such
conditions, when $NO_x$ levels are low, $NO_3$ mixing ratios may be sub-pptv and below the detection limit for most instruments
(Rinne et al., 2012).
The reaction of $NO_3$ with traces gases containing unsaturated C=C bonds proceeds via addition to form nitroxy-alkyl radicals
that undergo rapid reaction with $O_2$ to form nitroxy-alkyl peroxy radicals. The peroxy radicals react further (with $HO_2$, NO,
$NO_2$ or $NO_3$,) to form multi-functional organic nitrates, which can contribute to generation and growth of secondary organic
aerosols (Fry et al., 2014; Ng et al., 2016) or be lost by deposition.
The role of $NO_3$ as an oxidizing agent may be assessed via its total reactivity (or inverse lifetime). Whereas for OH,
experimental methods for measuring total reactivity in ambient air exist (Kovacs and Brune, 2001; Sinha et al., 2008), $NO_3$
reactivity has not yet been directly measured. Stationary-state approximations have often been used to calculate $NO_3$
lifetimes from its mixing ratio and production rate, the latter being given by: $k_1[NO_2][O_3]$ (Sobanski et al., 2016b; Brown et
al., 2009; Brown et al., 2007a; Brown et al., 2007b; Geyer and Platt, 2002; Heintz et al., 1996). Thus the stationary-state
turnover lifetime, $\tau_{ss}$,   can be calculated according to expression 1.
$$k_{ss} = \frac{1}{\tau_{ss}} = \frac{[NO_3]}{k_1[O_3][NO_2]}$$   (1)
This method is applicable when the chemical lifetime of $NO_3$ is sufficiently short so that stationary-state can be achieved
within transport time from emission to measurement location (Brown et al., 2003). Formally it is achieved when the
production and loss of $NO_3$ and $N_2O_5$ are balanced (Brown et al., 2003; Crowley et al., 2011). The time to acquire stationary-
state depends on production and loss rates for $NO_3$ and $N_2O_5$ and can take several hours. This approach can break down
under conditions of moderate to high $NO_2$ levels, strong sinks, low temperatures, or very clean air masses in which the sinks
for $NO_3$ and $N_2O_5$ become small (Brown et al., 2003). Indeed, Sobanski et al. (2016b) observed much lower stationary-state
loss rates compared to those calculated from measured VOC mixing ratios during the PARADE 2011 campaign and
concluded that this was mainly the result of sampling from a low lying residual layer with VOC emissions that were too
close for $NO_3$ concentrations to achieve stationary-state. They also considered the possibility that $NO_3$ may be formed by the
oxidation of $NO_2$ by Criegee Intermediates, which would bias calculations of its reactivity.
Summarizing, $NO_3$ reactivity with respect to gas-phase losses is a direct indication of night-time oxidation rates of VOCs,
with direct impacts on $NO_x$ levels by forming long-lived reservoir species (alkyl nitrates) some of which will partition to the
particle phase. Via modification of $N_2O_5$ concentrations, the $NO_3$ reactivity indirectly controls heterogeneous $NO_x$ losses and
$ClNO_2$ formation rates.
In this paper we describe a newly developed instrument that enables point measurements of $NO_3$ reactivity in ambient air.
After introducing the methodology in section 2, we show the results of extensive laboratory characterization of the



instrument along with discussion of the uncertainties associated with those measurements in section 3 to 5. In section 6 we
present a dataset of ambient $NO_3$ reactivity obtained at a forested / urban location in south-western Germany.
**2 Methodology**
Our experiments to measure $NO_3$ reactivity involve comparison of loss rates of synthetically generated $NO_3$ in zero-air and
in ambient air introduced into a flow-tube reactor. In zero-air, the loss of $NO_3$ is due to its reaction with $NO_2$ (present as a
necessary component in the generation of $NO_3$, see below) and losses on surfaces of the flow-tube. When zero-air is replaced
by ambient air, $NO_3$ is additionally removed by reaction with reactive gases present and its mixing ratio reduced accordingly.
An analysis of the change in signal for a fixed reaction time enables the $NO_3$ reactivity to be derived once certain corrections
have been applied (see below).
Figure 2 displays a schematic diagram of the experimental set-up. The three central components are a dark reactor for
generation of $NO_3$, the flow-tube in which $NO_3$ reacts with trace gases in ambient air samples and the detection system for
$NO_3$.
**2.1 Generation of $NO_3$**
Many laboratory studies of $NO_3$ kinetics have used the thermal decomposition of $N_2O_5$ as $NO_3$ source (R4) (Wayne et al.,
1991). The generation of $NO_3$ from gas-phase $N_2O_5$ eluted from samples of crystalline $N_2O_5$ (at -80 °C) was found to be
insufficiently stable for the present application and is also difficult to use during field campaigns where adequate laboratory
facilities for the safe generation and purification of $N_2O_5$ are frequently not available. In addition, this method of $NO_3$
generation was also accompanied by an $NO_2$ impurity of several parts per billion (ppbv).
We therefore generate $NO_3$ and $N_2O_5$ in situ, via the oxidation of $NO_2$ by $O_3$ (R1, R3). For this purpose, 400 standard $cm^3$
$min^{-1}$ (sccm) of synthetic air from a zero-air generator (Fuhr Cap 180) are passed over a Hg lamp (low-pressure, Penray type)
at a pressure of 1200 Torr. The photo-dissociation of $O_2$ at 184.95 nm results in formation of oxygen atoms that recombine
with $O_2$ to form ≈ 400 ppbv $O_3$. The $O_3$ / air flow is then mixed with $NO_2$ in synthetic air (0.93 ppbv, 1-10 sccm) and
directed into a temperature stabilized (30 °C), darkened, FEP coated reactor (length 70 cm, diameter 6 cm) also at a pressure
of 1200 Torr. The reactor is darkened to prevent the photolysis of $NO_3$ by room lights. Operation at above-ambient pressure
extends the reaction time for a given flow rate, thus optimising the conversion of $NO_2$ to $NO_3$ via the reaction between $NO_2$
and $O_3$, which has a low rate constant of $4.05 \times 10^{-17}$ $cm^3$ $molecule^{-1}$ $s^{-1}$ at 30 °C. The use of high pressures also optimises the
formation of $N_2O_5$ in the termolecular reaction R3, and reduces the rate of diffusion and loss of $NO_3$ to the walls of the
reactor. A high pressure in the darkened reactor also has the advantage of decoupling it from fluctuations in ambient
pressure, which influence the formation of $N_2O_5$. Heating the reactor to above room-temperature is carried out to stabilise the
formation of $N_2O_5$, which otherwise shows strong fluctuations owing to variations in laboratory temperature, typically about



3-5 degrees within the course of a day or night. The approximate reaction time for the stepwise conversion of $NO_2$ to $N_2O_5$ in
the darkened reactor is $\approx$ 5 min.
The gas exiting the darkened reactor passes through a pin-hole ($\emptyset \approx$ 250 µm) to reduce the pressure to roughly ambient level
and then enters a $\approx$ 30 cm long piece of ¼ inch ($\approx$ 6.4 mm) PFA tubing (residence time $\approx$ 0.5 s) which is heated to 140 °C in
order to thermally decompose $N_2O_5$ to $NO_3$. Calculations using the thermal decomposition rate constant for $N_2O_5$ (lifetime =
0.001 s at 140°C) indicate that after $\approx$ 0.1 s the $N_2O_5$ is stoichiometrically converted to $NO_3$. The temperature is measured on
the outside of the PFA tubing and does not necessarily reflect the temperature of the gas flowing through it. The value of
140 °C is chosen based on a series of experiments in which the tubing temperature was varied and the yield of $NO_3$
monitored. A PFA T-piece located immediately behind the heated tubing is used to add a 2900 sccm flow of either zero- or
ambient-air to the synthetic $NO_3$ sample. After this dilution step the air contains $\approx$ 50 pptv $NO_3$, $\approx$ 1 ppbv $NO_2$ and $\approx$ 50 ppbv
$O_3$. As described later, keeping $NO_2$ and $O_3$ levels as low as possible has important consequences for the data analysis. Low
levels of $NO_3$ also help to ensure that the addition of $NO_3$ does not significantly change the reactivity of the air, i.e. by
removing a large fraction of the reactive trace gases.
As described below, the present instrument is a modification of one designed to measure ambient mixing ratios of $NO_3$ and
$N_2O_5$ and is equipped with a second cavity connected to a heated inlet that measures the sum of $NO_3$ and $N_2O_5$. Experiments
in which both cavities were used to analyze the flow out of the heated piping indicated that there was no residual $N_2O_5$.

## 2.2 Detection of $NO_3$ using cavity-ring-down spectroscopy

For detection of the $NO_3$ radical we used Cavity Ring-Down Spectroscopy (CRDS), a sensitive technique for measurements
of atmospheric trace gases and often used for measurement of ambient $NO_3$ (Brown et al., 2002). In essence, CRDS is an
extinction measurement in a closed optical resonator (cavity) where light is trapped between mirrors with high reflectivity to
generate a very long absorption path. Ring-down refers to the decay of light intensity (monitored behind the cavity exit
mirror) and the general expression to derive the concentration of an absorbing or scattering gas is given by (Berden et al.,

135    2000):

$[X] = \frac{1}{c\sigma_{(x,\lambda)}}\left(\frac{1}{\tau_X} - \frac{1}{\tau_0}\right)$                                      (2)
Where $\tau_0$ and $\tau_x$ correspond to decay constants in the absence and presence of an absorbing or scattering trace gas X,
respectively and $\sigma_{(X,\lambda)}$ is the absorption cross-section / scattering coefficient of X at wavelength $\lambda$.
The instrument used is a two-channel CRDS that was previously used to measure ambient levels of $N_2O_5$ and $NO_3$ (Crowley
et al., 2010; Schuster et al., 2009). Important modifications to the previous set-up include use of FEP coated glass cavities of
equivalent size and fibre-optics for the coupling of the laser to the cavity. The thermal dissociation cavity previously used for
detection of atmospheric $N_2O_5$ is not necessary for the measurement of $NO_3$ lifetimes but was used for calibration and
characterization experiments. Only the central features and important modifications compared to the prototype described in
Schuster et al., (2009) are described in detail here.





The light source is a 625 Hz, square-wave modulated, 100 mW laser-diode located in a Thor Labs TCLDM9 housing and
thermally stabilized at 36°C using a Thor Labs ITC 502 Laser-Diode Combi Controller to produce light at 661.95 nm (0.5
nm full width at half maximum) and therefore close to the $NO_3$ absorption maximum. The effective cross section of $NO_3$ was
calculated as 2.09 x $10^{-17}$ $cm^2$ $molecule^{-1}$ by convoluting the temperature dependent $NO_3$ absorption spectrum (Orphal et al.,
2003; Osthoff et al., 2007) with the laser-diode emission spectrum. Coupling between the laser-diode and the cavities is
achieved by using either optical fibres (0.22 NA, 50 µm core, 400-2400 nm) for measuring $NO_3$ reactivity or using fibre-
optics with a beam splitter (Thor Labs FCMM50-50A-FC, 50:50 ratio) in order to operate both cavities. The beam was
collimated (Thor Labs FiberPort Collimator PAF-X-18-PC-A) and directed through an optical isolator (Thorlabs IO-3D-660-
VLP), focused by a lens (Thorlabs A230TM-A) into the optical-fibre and then collimated again to a beam diameter of about
6 mm before entering the cavity.
The $NO_3$ cavity (Teflon-coated glass (DuPont, FEP, TE 9568), length 70 cm, volume 79$cm^3$) was operated at room
temperature, while the $N_2O_5$ cavity was operated at 80°C with a pre-cavity section heated to 85°C in order to convert $N_2O_5$ to
$NO_3$. The $NO_3$-cavity was connected to the flow-tube using 1/8" (≈ 3.2 mm) PFA tubing that lined the 1/4" (≈ 6.4 mm)
injector. The use of small diameter tubing results in short transport times between the flow-tube and CRDS and also induces
a pressure drop of 133 mbar, so that the pressure in the cavity was 880 mbar. Gases entered the middle of the cavity via a T-
piece and were pumped from the ends via a flow controller into the exhaust. The flow rates in both cavities were 3000 $cm^3$
(STP) $min^{-1}$ (sccm) resulting in a residence time of approximately 1.6 s as calculated from the volume flow. Gas entering the
CRDS detector was always passed through a 2 µm membrane filter (Pall Teflo) to remove particles. Light exiting the cavities
through the rear mirror was detected by a photomultiplier (Hamamatsu E717-500) which was screened by a 662 nm
interference filter. The pre-amplified PMT signal was digitized and averaged with a 10 MHz, 12 bit USB scope (Picoscope
3424) which was triggered at the laser modulation frequency of 625 Hz.
The ring-down constant in the absence of $NO_3$ was obtained by adding NO (1-3 sccm of a 100 ppmv mixture NO in $N_2$)
every 40 points of measurement for approximately 15 s. Titration with NO took place at the inlet of the T-shaped glass
cavity giving the gas mixture sufficient time to react with $NO_3$. The $L/d$ ratio (the ratio of the distance between the cavity
mirrors ($L$) and the length of the cavity that is filled by absorber ($d$)) was determined as described previously (Schuster et al.,
2009; Crowley et al., 2010) and was 1.01 ±0.03. Values of $\tau_0$ in dry zero-air at 760 Torr were usually between 140 and
160 µs indicating optical path lengths of ≈ 42-48 km. When operated at a flow of 3000 sccm, the noise levels on the $NO_3$
signal are such that the precision (3s integration interval) is better than 1 pptv. As we describe later, the $NO_3$ reactivity is
derived from measurements of the relative change in the $NO_3$ mixing ratio, so that the precision rather than total uncertainty
in the $NO_3$ mixing ratio defines the accuracy of the reactivity measurement.



### 2.3 Flow-tube for NO₃ reactivity measurement

The flow-tube, thermostatted to 20 °C by flowing water through an outer jacket, is an FEP-coated glass tube of length 50 cm and internal diameter 4 cm. Gas enters the flow-tube at one end via a conical section with a 3/8 inch ($\approx$ 9.5 mm) glass fitting through which ¼ inch ($\approx$ 6.4 mm) PFA tubing could be inserted. The total flow through the flow-tube was 3300 sccm, consisting of 400 sccm from the darkened reactor and 2900 sccm zero-air / ambient air. The flow and pressures indicated above, result in a Reynolds number of $\approx$ 123 (i.e. laminar flow) in the cylindrical part of the flow-tube, but with an entrance length (Le) to acquire laminar flow of 27 cm indicating that the flow-tube operates in a mixed turbulent / laminar flow regime.

$$Le = 0.112 \, r \, Re \tag{3}$$

Gases exit the flow-tube via a length of 1/8 inch PFA tubing supported in an axially centred stainless-steel tube (length 50 cm, diameter ¼ inch) which could be translated along the major flow-tube axis thus changing the contact (reaction) time between NO₃ and any reactive species or the flow-tube wall. In principal, this enables the dynamic range of the measurement to be adjusted (i.e. long contact times for low reactivity, short contact times for high reactivity) though we found that reactivity-dependent dilution of the ambient air was a better method to extend the dynamic range to high reactivities as very short reaction times were not possible due to a finite residence time in the CRDS detection system and also due to mixing effects in the flow-tube. In order to prevent formation of a "dead volume" at the back of the flow-tube beyond the tip of the outlet, 400 sccm were removed via a critical orifice to the exhaust pump. During measurement of NO₃ reactivity the extraction point was usually set for a reaction time of about 10.5 s, which was determined as described below.

As described later, to derive the NO₃ reactivity we compare its concentration in zero-air to that in ambient air samples. We found that when switching between sampling ambient air and dry, zero-air, the resulting change in relative humidity caused an abrupt change in NO₃ which then slowly recovered towards its original value. Measurement of the wall loss rate of NO₃ in dry and humidified zero-air by moving the injector (see below) revealed no substantial difference and we conclude that the change in NO₃ is due to wall loss at the point of mixing of NO₃ flows and the zero-air flow, which is very turbulent. In order to eliminate data loss while waiting for signals to stabilise following zeroing, we humidify the zero-air to the same absolute humidity (±2 %) as ambient. To do this, the ambient relative humidity was monitored by passing 100 sccm air over a sensor that recorded both temperature and relative humidity. The zero-air was humidified by directing a variable fraction of the (constant) total flow through a 2 l gas wash-bottle filled with HPLC grade water. The relative humidity of the resulting mixture was matched to ambient levels by dynamic adjustment of the fractional flow passing through the wash-bottle. The zero-air used for purging the mirrors as well that used for NO₃ generation was not humidified.

In order to ensure that air from the zero-air generator was free of reactive gases that survived the catalytic purification process, we compared it to hydrocarbon-free, bottled synthetic air (Westfalen). No change in the concentration of [NO₃] could be observed when switching between zero-air and bottled air, indicating that the zero-air generator was suitable.



However, poisoning of the catalyst of the zero-air generator by amines, sulphides or thiols or contamination of the filters
could potentially become problematic when using compressed, highly polluted ambient air.
**2.3.1 Derivation of the effective reaction time and wall loss rate constant for $NO_3$**
In flow-tubes where radial, diffusive mixing of gases is rapid (i.e. at low pressures of He and "plug-flow" conditions), the
effective reaction time can be close to that calculated from the volumetric flow rate once axial diffusion is accounted for
(Howard, 1979). At higher pressures and laminar flow, reactions times are defined by the parabolic velocity distribution and
extent of radial mixing whereas high pressure flow-tubes operated under turbulent conditions (Reynold numbers > 3000)
plug-flow can be achieved (Donahue et al., 1996; Seeley et al., 1993). According to the calculations of Reynolds numbers
outlined above, our flow-tube is not operated in either a pure laminar or turbulent regime, which can make accurate
calculation of the reaction time difficult. Using the volumetric flow rate and flow-tube diameter, we calculate an average,
linear velocity of the gas of 4.78 cm s$^{-1}$ at 760 Torr and 298 K in the cylindrical section of the flow-tube. This enables us to
calculate the injector position dependent reaction time in the flow-tube, which for 45 cm is 9.5 s. This should be regarded as
an initial estimate of the true reaction time as it does not consider the non cylindrical section of the flow-tube (2.5 % of total
volume), the radial distribution of velocities in the flow-tube or mixing effects. A further additional 1.6 s must be added to
this to take the average reaction time in the cavity into account (calculated from the cavity volume and the flowrate) resulting
in an approximate, total reaction time of 11.1 s.
A further method to derive an "effective" or averaged reaction time is to add a short pulse of gas to the flow-tube and
monitor its arrival time at the detector. However, as $NO_3$ cannot be easily stored, we instead add a pulse of a reactant that
removes $NO_3$. A syringe was therefore used to add a short pulse (0.1 cm$^3$ in < 0.5 s) of NO diluted in $N_2$ (0.22 ppbv) to the
flow-tube at the T-piece where the $NO_3$ source and zero-air are mixed.
The resultant depletion in the $NO_3$ signal (measured at a time resolution of 0.35 s) displayed an inverted Gaussian form with
an elongated flank after the minimum (Fig. 3) which can be attributed to non-isothermal effects, secondary flows and
recirculation processes in the flow-tube (Huang et al., 2016) which require fluid dynamics simulations to be fully
characterised. The average reaction time, $t$, can however be derived from:
$t = \frac{\sum I_j t_j}{\sum I_j}$ (4)
where $I_j$ is the signal recorded at each time step $t_j$
In total, 25 experiments were conducted, resulting in an effective reaction time of 11.4 ± 0.5 s determined via expression (4).
The two methods outlined above thus provide approximate values for the reaction time which are in good agreement (< 3 %
deviation).
As the reaction time is a central parameter for calculating the $NO_3$ reactivity, a third method was employed, in which a
known amount of NO was added at the usual mixing point and the depletion in $NO_3$ observed. As the rate constant for
reaction of NO with $NO_3$ is known with an uncertainty (at room temperature) of 13 %, this should enable derivation of an





effective reaction time that also takes all mixing effects (both in the flow-tube and cavity) into account. In a series of
experiments, known amounts NO were added to the 2900 sccm flow of zero-air (via a calibrated mass flow controller) at the
usual mixing point. In the absence of other processes which remove or form $NO_3$, its change in concentration upon adding
NO is described by:
$[NO_3]_t = [NO_3]_0 \, exp^{-(k_2[NO]+k_w+k_3[NO_2])t}$ (5)
Where $[NO_3]_0$ and $[NO_3]_t$ are the concentrations of $NO_3$ before and after addition of NO, respectively. $k_2$ and $k_3$ are the rate
constants for reaction of $NO_3$ with NO and $NO_2$, respectively at the flow-tube / cavity temperature, $k_w$ is the rate constant (s⁻
¹) for loss of $NO_3$ at the flow-tube walls and $t$ is the desired parameter. Rearranging, we get a simple expression (6), which
shows that a plot of $ln([NO_3]_t$ versus [NO] should yield a slope of $k_2t$, from which $t$ can be derived using an evaluated and
recommended value of $k_2$ (Atkinson et al., 2004). Once corrected for the contribution from $k_5[NO_2]$, the intercept should, in
principal, give a value of $k_w$.
$ln\frac{[NO_3]_0}{[NO_3]_t} = k_2[NO]t + k_w + k_3[NO_2]$ (6)
A plot of $[NO_3]_t$ versus [NO] is displayed in Fig. 4a for three different amounts of added $NO_2$. Although the curve follows
roughly exponential behaviour as expected, the slopes and thus the value of $t$ obtained was found to depend on the initial
$NO_2$ concentration, with values of 5.7, 5.1 and 4.5 s obtained for $NO_2$ mixing ratios of 2.94, 5.88 and 8.82 ppbv,
respectively. This indicates that the kinetics of $NO_3$ formation and loss are more complex than defined by expression (6) and
the relative rates of reaction of $NO_3$ with NO (R2) and $NO_2$ (R3) and its formation via $N_2O_5$ decomposition (R4) and reaction
of $O_3$ with $NO_2$ (R1) in the flow-tube all impact on the $NO_3$ mixing ratio. In Fig. 4b we display the results of a similar
experiment in which $NO_2$ as added. In this case, there is obvious curvature in the plot of $[NO_3]_t$ versus $[NO_2]$, which is not
predicted by expression (5). The decomposition of $N_2O_5$ formed by reaction R3 as well as oxidation of $NO_2$ by $O_3$ (R1, see
section 3.1) both lead to the formation of $NO_3$ and are the causes of this behaviour, especially at high $[NO_2]$ and low [NO].
At the flow-tube and cavity temperature (circa 298 K), the rate constant for decomposition of $N_2O_5$ ($k_4$) is $4.4x10^{-2}$ s⁻¹
(Atkinson et al., 2004).
Extraction of the reaction time thus required numerical simulation of the data obtained by adding various amounts of NO, to
the flow-tube in the presence of different $NO_3$ and $NO_2$ concentrations. The impact of reactions R2, R3 and R4 was assessed
by numerical simulations using FACSIMILE (Curtis and Sweetenham, 1987) and considering the reactions listed in Table 1.
The input parameters for the simulations were the concentrations of NO, $NO_2$ and $O_3$ and the rate constants, which were
taken from IUPAC recommendations (Atkinson et al., 2004). The total reaction time ($t$) and the wall-loss rate constant for
$NO_3$ ($k_w$) were adjusted until each of the six datasets could be reproduced with a single value for each parameter. The initial
concentration of $[NO_3]_0$, was allowed to float until best agreement was achieved. This way, the reaction time was determined
to be 10.5 s, which is in good agreement with that derived by pulsed addition of NO. As our reactivity derivation relies on
the change in $NO_3$ signal upon adding a reactant to the flow-tube, we consider the value of 10.5 s, which takes mixing,





diffusion etc. into account to be the most appropriate value but assign an uncertainty (± 1 s) that overlaps with the other
methods. The wall loss rate of $NO_3$ (which is independent of the NO and $NO_2$ concentrations) was found to be $4 \times 10^{-3}$ $s^{-1}$.
For analysis of ambient reactivity we use a reaction time of 10.5 s as derived from the addition of NO. This means that our
ambient reactivities are directly tied to the rate constant for reaction between $NO_3$ and NO. As described later, during
ambient measurements we periodically add a known amount of NO to the zero-air to monitor a known reactivity under real
operating conditions.
Figures 5a and 5b show the correlation between simulated and measured $NO_3$ concentrations in these experiments. In both
cases the slope is close to unity (0.97-1.02) with an intercept close to zero. A set of similar experiments performed at 30 %
and 80 % humidity also showed excellent agreement using the same values of $t$ and $k_w$. We conclude that the behaviour of
$NO_3$ in this system can be very accurately predicted by numerical simulations using a simple reaction scheme under a variety
of conditions (initial $NO_3$, NO and $NO_2$ varied), giving us confidence in our ability to extract loss rates for $NO_3$ in ambient
air.
When gas-phase reactivity is low, a substantial fraction of $NO_3$ may be lost via collisions with the walls rather than due to
reactive gases. For this reason, we re-measured the value of $k_w$ obtained above in a further set of experiments in which the
$NO_3$ concentration was measured as a function of injector position (contact time in the flow-tube) at a constant initial mixing
ratio of $NO_3$ and $NO_2$ and in the absence of NO. For this we calculate the reaction time for each of the three injector
positions from pulsed addition of NO as described above, but normalized to the reaction time derived from addition of NO
with numerical simulation. The results of such an experiment are displayed in Fig. 6 and we draw attention to the fact that,
even at maximum reaction time (10.5 s), the change in the $NO_3$ concentration is only about 10 %. This reflects the low
efficiency of reaction of $NO_3$ with the FEP coated glass walls. To put this result in context, performing the same experiment
in a non-coated glass tube results in the almost complete loss of $NO_3$. The numerical simulation was initialised with the same
set of rate parameters described above, a fixed $NO_2$ concentration and only $k_w$ and the initial $NO_3$ concentration were varied.
The best fit was obtained when $k_w$ was $4 \times 10^{-3}$ $s^{-1}$, in agreement with the simulations at fixed time and variable NO and $NO_2$.
Using expression (7), this value of $k_w$ can be converted to an approximate uptake coefficient for $NO_3$ to the FEP-coated tube
of $\approx 5 \times 10^{-7}$.
$$\gamma = \frac{2\,r\,k_w}{c} \qquad\qquad (7)$$

**3 Data analysis and derivation of $NO_3$ reactivity**

We first consider the passage of $NO_3$ through the flow-tube in a flow of zero-air. If $NO_3$ is lost in one or more pseudo-first-
order processes, its decay should be exponential and its concentration, $[NO_3]_t^{ZA}$ after a reaction time $t$, is given by expression

301    (8).

$$[NO_3]_t^{ZA} = [NO_3]_0^{ZA} exp^{(-k_{ZA}t)} \qquad\qquad (8)$$



Where the superscript "ZA" refers to use of zero-air. As $NO_3$ is lost only via reaction with $NO_2$ and to the wall, $k_{ZA} =$
$k_{wall} + k_{NO2}$ where $k_w$ is the first-order loss rate constant for wall-loss and $k_{NO2}$ is the first-order loss rate constant for
reaction with $NO_2$ and is equal to $k_3[NO_2]$. When zero-air is switched for ambient air containing reactive trace gases (RTG),
we have:
$[NO_3]_t^{Amb} = [NO_3]_0^{Amb} exp^{(-k_{Amb}t)}$ (9)
where $k_{Amb} = k_w + k_{NO2} + k_{RTG}$ and $k_{RTG}$ is the first-order loss rate constant for reaction of $NO_3$ with trace gases present
in ambient air other than $NO_2$.
If $[NO_3]_0^{ZA}$ and $[NO_3]_0^{Amb}$ are equivalent, expression 10 is obtained.
$\dfrac{[NO_3]_t^{ZA}}{exp^{(-k_{ZA}t)}} = \dfrac{[NO_3]_t^{Amb}}{exp^{(-k_{Amb}t)}}$ (10)
Rearranging and substituting for $k_{ZA}$ and $k_{Amb}$ leads to
$k_{RTG} = \dfrac{\ln\left(\frac{[NO_3]_t^{ZA}}{[NO_3]_t^{Amb}}\right)}{t} = \frac{1}{\tau}$ (11)
Where $\tau$ is the $NO_3$ lifetime. In principal, it should thus be possible to calculate the reactivity of $NO_3$ in ambient air by
measuring $[NO_3]_t^{ZA}$, $[NO_3]_t^{Amb}$ and knowing the reaction time $t$. Later we discuss the applicability of this expression and
show that corrections are necessary to take the re-formation of $NO_3$ into account, especially when dealing with air-masses
with high $NO_2$ content. This is similar to the laboratory experiments described above and required numerical simulation,
which we present below.
The concentration of $NO_3$ in zero-air measured when the injector is positioned for maximum reaction time, $[NO_3]_t^{ZA}$, was
measured by flushing the inlet with 3000 sccm zero-air creating an overflow of $\approx$ 100 sccm. When switching to ambient
measurements, the zero-air overflow was redirected via a flow controller, $F_3$, that connected the zero-air overflow line to the
exhaust and which was set to 3500 sccm. This setup has the advantage of enabling dynamic dilution of ambient air. If the
reactivity is so high that the $NO_3$ levels approached the detection limit, $F_3$ does not withdraw the entire 3500 sccm overflow
but allows e.g. 2000 sccm to be added to the inlet, resulting in sampling 900 sccm of ambient air plus 2000 sccm of zero-air,
a dilution factor of 2900/900 which is slightly increased by the 400 sccm flow from the darkened reactor. A five point
dynamic dilution with zero-air is implemented in the software, which changes the set point for $F_3$ and dilutes the ambient air
with zero-air if $[NO_3]_t^{Amb}$ decreases below 10 pptv for an average time period of 30 s. Conversely, the dilution can be
decreased again if $[NO_3]_t^{Amb}$ becomes $\geq [NO_3]_t^{ZA} - 10$ ppt. Dilution factors ($D_i$) were determined using a Gilibrator flow
meter (Gilian Gilibrator-2) and were: $D_1$=1.14 for the measurement of pure ambient air (here the small dilution effect is
caused by the 400 sccm zero-air used in the production of $NO_3$), $D_2$=1.74, $D_3$=3.71, $D_4$=8.98, $D_5$=14.07 when diluting
ambient air. With increasing dilution, errors in the measurement will increase as well (see later).
The analytical expression given above to derive the $NO_3$ reactivity is an ideal case in which $NO_3$ is lost by a number of first-
order processes and is not formed in the flow-tube to a significant extent. However, as we already demonstrated in the
laboratory experiments to examine the effects of varying NO, $NO_2$ and $NO_3$ concentrations, the formation of $N_2O_5$ in the



reaction of $NO_3$ with $NO_2$ (R3) and its thermal decomposition back to $NO_3$ can impact on the $NO_3$ concentration as $NO_2$ is
present both in the mixture used to generate $N_2O_5$ and $NO_3$ and also in ambient air. While the formation of $N_2O_5$ from $NO_2$
and $NO_3$ (R2) is, to a good approximation, independent of temperature between about 280 and 305 K, the rate constant for
thermal decomposition of $N_2O_5$ (R3) varies by a factor of 26 over the same temperature range. The simple, analytical
approach outlined above thus fails at temperatures where the decomposition of $N_2O_5$ is important and when sufficient $NO_2$ is
present to account for a significant fraction of the loss of $NO_3$. This is illustrated In Fig. 7a, in which simulations of the $NO_3$
concentration at a reaction time of 10.5 s and at different temperatures and amounts of $NO_2$ (as reactant) are displayed and
compared with the simple exponential behaviour (black data points) calculated from expression (11). The simulations show
that the dependence of the $NO_3$ concentration on $NO_2$ is non-exponential, indicating that re-generation of $NO_3$ from the $N_2O_5$
formed is significant, especially at higher temperatures. Figure 7b plots the ratio of the true reactivity (i.e. that used as input
into the numerical simulation) versus that obtained by analyzing the simultaneous change in $NO_3$ concentration using
expression (11). It is evident that the use of this expression generally results in underestimation of the true reactivity due to
the formation and decomposition of $N_2O_5$. The bias will be largest when sampling polluted air where the reactivity has a
large component due to $NO_2$ and small under conditions of low $NO_2$ and high $k_{RTG}$ typical for remote, forested areas.
However as previously mentioned the decomposition of $N_2O_5$ is strongly temperature dependent so that the bias will increase
with rising temperature and decrease with sinking flow-tube temperature.
Apart from the formation and thermal dissociation of $N_2O_5$, the reaction of $NO_2$ with $O_3$ may, under some conditions,
represent a further potential source of $NO_3$ in the flow-tube despite the low rate constant for (R1). Due to the in-situ method
of production of $N_2O_5$ and $NO_3$ in the dark reactor, $NO_2$ (0.6-3 ppbv) and $O_3$ (40-50 ppbv) are always present in the flow-
tube. $NO_3$ generated in the flow-tube was therefore simulated for different amounts of $O_3$ and $NO_2$ corresponding to the
minimum and maximum mixing ratios used in our experiments. Figure 8 indicates that with 50 ppbv of $O_3$ and 2 ppbv $NO_2$,
< 0.5 pptv of $NO_3$ is formed in the 10.5 s available for reaction in the flow-tube, which would not strongly impact on the
results if the analytical expressions above were used to derive the $NO_3$ reactivity. Under highly polluted conditions (e.g. 100
ppbv $O_3$ and 20 ppbv $NO_2$) the effect is however measureable (> 2 pptv).
The discussion above indicates that the use of expression (11) can, under certain circumstances (e.g. low $NO_x$, high $NO_3$
reactivity to VOCs) give a reasonable representation of the $NO_3$ reactivity. However, in order to be able to derive $NO_3$
reactivities from any air mass we prefer to use numerical simulation take $NO_3$ reformation into account and enable extraction
of accurate values in any conditions.

### 3.1 Numerical simulations for extraction of ambient reactivity

In this section we outline the experimental procedure and the associated data analysis for extracting the $NO_3$ reactivity from
an ambient dataset as exemplified by the data shown in Fig. 9. This data covers a 1 hour period in which several phases of
inlet-overfilling with humidified zero-air and titration with NO are apparent as are periods of mixing $NO_3$ with ambient air.



The dataset has already been corrected for baseline drift in the $NO_3$ zero during titration, hence each titration-zero is
scattered around 0 pptv $NO_3$.
The periods marked "ZA" (zero-air) were used to extract the $NO_3$ concentration after a residence time of 10.5 s in flow-tube
in the absence of ambient reactive trace gases. The data show that a plateau in the $NO_3$ signal with zero-air is observed after
about 2-3 titration cycles are complete, which is the result of slow flushing through the inlet of reactive gases. Once a stable
signal is acquired, $[NO_3]_{t=10.5}^{ZA}$ can be taken as an average value for each 300 s zero-air phase. These values are then used to
calculate the initial $NO_3$ concentration $[NO_3]_0^{ZA}$, i.e. before $NO_3$ enters the flow-tube. This was done in an interative
procedure using numerical simulation with FACSIMILE embedded in a separate program. Input values are the $O_3$ and $NO_2$
concentration (from the darkened reactor), a first estimate for $[NO_3]_{t=0}^{ZA}$ and the rate coefficients for the $NO_3$ reactions listed
in Table 1. At the end of the simulation (a few seconds of computing time) the simulated and measured values of $[NO_3]_{t=10.5}^{ZA}$
are compared and the ratio used to adjust the next input value for $[NO_3]_{t=0}^{ZA}$. The iteration continued until convergence was
reached. Convergence was considered satisfactory when the deviation between measured and simulated values of
$[NO_3]_{t=10.5}^{ZA}$ was less ≤1 %. This usually took only 5 simulations per data point as the initial value for each new time point
was chosen to be the final value for the preceding time point. Ideally, $[NO_3]_{t=0}^{ZA}$ should be constant over long periods of time.
In fact, deviations of several pptv, especially during field measurements, were observed over periods of hours and so values
of $[NO_3]_{t=0}^{ZA}$ were linearly interpolated to each time point in which ambient reactivity was recorded.
Once initial $NO_3$ concentrations had thus been obtained a new set of simulations was started to simulate the measured values
of $[NO_3]_{t=10.5}^{Amb}$ . In this case, the simulation was initialized with the values of $[NO_3]_{t=0}^{ZA}$ obtained as described above and the
total $NO_2$ concentration and $O_3$ concentrations, which contained a constant contribution from the dark-reactor and a variable
concentration from ambient $NO_2$ and $O_3$ once corrected by the dilution factor (see above). An initial estimate of the total
$NO_3$ reactivity, $k_{RTG}$, was made and the simulated value of $[NO_3]_{t=10.5}^{Amb}$ compared to that measured. The simulation was
iterated, with incremental adjustment of $k_{RTG}$ until agreement between simulation was ≤1 %. For ambient datasets, in which
the reactivity can be highly variable this sometimes took several iterations, though as each simulation took less than a second
this is not a particularly time consuming procedure.
**4 Reactivity of an isoprene standard.**
To validate our experimental and analytical procedure, we performed reactivity measurements on a bottled isoprene standard
(0.933 ± 0.09 ppmv, Westfalen), diluted in zero-air. Isoprene was chosen as it is an important biogenic reactant for $NO_3$ in
the troposphere and also because the rate coefficient, $k_{isoprene}$, for its reaction with $NO_3$ has been studied on many occasions
(Atkinson et al., 2006; IUPAC, 2016) and therefore has a low associated uncertainty ($k_{isoprene}$ = 6.5 ± 0.15 x $10^{-13}$ cm$^3$
molecule$^{-1}$ s$^{-1}$ at 298 K).



Experiments were carried out at various isoprene and $NO_2$ mixing ratios and the results are summarized in Fig. 10, which
indicates excellent agreement between the measured reactivity and that calculated from the isoprene mixing ratio and rate
coefficient, the slope of an unweighted fit being $1.00 \pm 0.03$. The error bars on the calculated reactivity represent total
uncertainty in the isoprene and $NO_2$ mixing ratio, the reaction time and the rate coefficient. These results confirm that the
instrument and data analysis procedure measure accurate values of $NO_3$ reactivity in the presence of $NO_2$ and organic
reactants.

**5 Detection Limit, dynamic range and overall uncertainty**

While the overall uncertainty associated with absolute $NO_3$ concentration measurement are influenced by factors such as
uncertainty in the cross-section as well as in the measurement of the laser emission spectrum the fractional change in
concentration used to derive the $NO_3$ reactivity is not impacted. The detection limit for measuring $NO_3$ reactivity is defined
by the minimal detectable change ($MDC_{NO3}$) in the $NO_3$ mixing ratio. This depends on noise levels and drift in ring-down-
time, i.e. on the precision of the $NO_3$ signal and also on the stability of the synthetically generated $NO_3$. The instrumental
noise on the $NO_3$ signal was reduced by averaging over $\approx 3$ s per data-point ($\approx 1800$ ring-down-events) to give a noise
limited detection limit of $\approx 0.2$ pptv. Precision is limited by the stability of the CRDS setup where changes in the mirror
reflectivity induced by thermal or mechanical stress can lead to a drift in the ring-down time. The precision can be estimated
from the standard deviation of the signal from one zeroing period to the next over the measurement period. Under typical
laboratory conditions this was normally $\approx 0.7$ pptv.
Since $[NO_3]_0^{ZA}$ is interpolated onto the measured $[NO_3]_t^{Amb}$ time series to calculate the reactivity, the stability of the $NO_3$
source is of great importance. Changes in the amount of synthetically generated $NO_3$ are caused by fluctuations in the
temperature or pressure of the dark-reactor, the flow of $NO_2$ and changes in the intensity of light from the $O_3$ generator. In
general, the poorer the stability of the $NO_3$ source chemistry, the more frequently the $NO_3$ mixing ratio in zero-air has to be
measured. In laboratory conditions, changes of $\pm 1$ pptv within one hour were typical, making $[NO_3]_t^{ZA}$ measurements every
1200 s more than sufficient. In field conditions, where the instrument housing may be subject to larger temperature
fluctuations, more frequent determination of $[NO_3]_t^{ZA}$ may be necessary. The $NO_3$ source stability was obtained from the
standard deviation of the averaged $[NO_3]_t^{ZA}$ concentrations and propagating this with the standard deviation of two
consecutive $[NO_3]_t^{ZA}$ measurements, for which typical values in laboratory conditions were $\approx 1$ ppt. To define an overall,
minimal detectable change in $NO_3$ ($MDC_{NO3}$), the noise and drift limited precision was combined with the $NO_3$ source
stability to result in $MDC_{NO3} = 2.5$ pptv.
An $MDC_{NO3}$ of 2.5 pptv results in a lower limit for the measurement of $NO_3$ reactivity of 0.005 s$^{-1}$ (obtained from expression
(11) with $[NO_3]_t^{ZA} = 50$ pptv and $[NO_3]_t^{Amb} = [NO_3]_t^{ZA} - MDC_{NO3} = 47.5$ pptv, at the lowest dilution factor of 1.14). An
upper limit for the measurable reactivity is 45 s$^{-1}$, largely defined by the uncertainty of the dilution factor. Dilution factors



were obtained by measurements of the actual flows going into the flow-tube using a Gilibrator flow meter (Gilian Gilibrator-
2, stated accuracy ±1%). The total uncertainty in the dilution factor is defined by the accuracy of the measurement of the
dilution flows as well as by the accuracy of the flow controllers used for flow regulation (± 2 %) and was calculated to be
2.5 %. The error in the calculated reactivity is lowest for the lowest dilution but if the $[NO_3]_t^{Amb}$ gets close to the detection
limit this will also have a strong influence on the calculated reactivities making a higher dilution factor favourable. Dilution
factors were chosen to keep the instrument operating in a region (10 pptv < $NO_3$ < 40 pptv) where both effects are
minimized.
A minimum detectable change in $NO_3$ of 2.5 pptv leads to an uncertainty of ≈ 15 %, when $NO_3$ varies between ≈ 10 und 30
pptv (starting from 50 pptv in zero-air). The uncertainty increases dramatically when $NO_3$ levels are close to 50 pptv (i.e.
very low reactivity) or less than 5 pptv (very high reactivity without dilution). This is illustrated in Figure S1 of the
supplementary information. As mentioned in section 2.3.1 the uncertainty in the reaction time (10 %) also contributes to the
overall uncertainty.
To assess the uncertainty associated with derivation of the $NO_3$-reactivity from numerical simulation, uncertainties
associated with the input parameters have to be considered. As previously demonstrated (Groß et al., 2014) this is best
assessed in a Monte-Carlo approach in which the key parameters are varied within a range reflecting their uncertainty limits.
The parameters that most sensitively influence the derived value of $NO_3$ reactivity are the $NO_2$ mixing ratio and the rate
coefficients for $N_2O_5$ formation ($k_3 = 1.2 \pm 0.1 \times 10^{-12}$ cm$^3$ molecule$^{-1}$ s$^{-1}$) and decomposition ($k_4 = 4.4 \pm 0.4 \times 10^{-2}$ cm$^3$
molecule$^{-1}$ s$^{-1}$). The rate coefficients listed are for 1 bar and room temperature as appropriate for the experimental conditions,
the uncertainties quoted (≈ 10 %) are based on assessment of kinetic data (Burkholder et al., 2016). The Monte Carlo
simulations were initiated with a $NO_3$ mixing ratio (in zero-air) of 50 pptv, decreasing to 20 pptv upon reaction with air. In
total, 6 sets of ≈ 1200 simulations were carried with variation of the initial $NO_2$ mixing ratio between 1 and 5 ppbv and the
associated error in $NO_2$ mixing ratio was taken as 8 %. For any given simulation, the output value of the $NO_3$ reactivity
($k_{RTG}$) was stored. The 2σ uncertainty was derived from the Gausssian fits to histograms of $k_{RTG}$ (insets at $NO_2$ = 1.0, 3.0 and
5.0 ppbv) and is plotted (as a percent of $k_{RTG}$) versus $k_{RTG}$ / $NO_2$. The latter may be considered a measure of whether $NO_3$
reacts predominantly with $NO_2$ to form $N_2O_5$ ($k_3$) or with reactive trace gases. Fig. 11 shows that the uncertainty associated
with the simulations is very sensitive to ambient $NO_2$ levels, varying between > 100 % (at 5 ppbv $NO_2$ and a reactivity of
0.017 s$^{-1}$) to 3.1 % (at 1 ppbv $NO_2$ and a reactivity of 0.092 s$^{-1}$) of the extracted $k_{RTG}$. Clearly, the extraction of $k_{RTG}$ is most
accurate in conditions of low $NO_x$ and when $NO_3$ lifetimes are short (e.g. forested regions far from anthropogenic activity).
Another potential bias in the measurement is the temperature dependence of the rate constant of the reactions of trace gases
with $NO_3$. Measurements were normally conducted at 20 °C in the flow-tube whilst outside temperature can differ from this.
However, (unlike OH) the $NO_3$ reactions which dominate its reactivity involve addition to double bonds (e.g. of terpenes)
and are only weakly temperature dependent. Therefore, to a good approximation, this error can be neglected. Under



circumstances where the reactivity is known to be driven by reaction with reactive trace gases for which $NO_3$ has large
temperature dependence this error has to be taken into consideration.
The overall uncertainty thus derives from a combination of measurement errors (cavity instability, drift in $NO_3$ source etc.)
and the need to correct for $NO_3$ reactions with $NO_2$. Under ideal conditions (e.g. as described above for laboratory operation)
the former can be reduced to $\approx$ 16 %. For a scenario in which biogenic VOCs dominate $NO_3$ reactivity in a low $NO_x$
(< 1 ppbv) environment an additional uncertainty of $\approx$ 6-10 % from the numerical simulations results in a total uncertainty of
$\approx$ 17-20 %. In a high $NO_x$ environment, the total uncertainty will be dominated by that associated with the simulations. For
example, at 5 ppbv $NO_2$ and a reactivity of 0.03 $s^{-1}$ the total error would be close to 45-50 %.

## 6 Deployment in the NOTOMO campaign, 2015

The $NO_3$ reactivity set-up described above was deployed for the first time in the field during the NOTOMO campaign
(NOcturnal chemistry at the Taunus Observatorium: insights into Mechanisms of Oxidation) in the Taunus mountains (S.W.
Germany) in 2016. The site, previously described in detail (Crowley et al., 2010; Sobanski et al., 2016b), is situated on top of
the "Kleiner Feldberg" mountain (850 m above sea level) in a forested area with urban influence. The site is impacted by
biogenic emissions from forested regions (mainly in the north/west) and by anthropogenic emissions from the local urban
centres of Frankfurt, Mainz and Wiesbaden in the south-east to south-west.

### 6.1 Reactivity measurements during NOTOMO

The $NO_3$ reactivity instrument was located in a research container and sampled from a common, high-flow inlet together
with other instruments. The high-flow inlet was driven by an industrial fan drawing 10 $m^3$ $min^{-1}$ through a 15 cm diameter
stainless steel pipe with its opening about 8 m above the ground. This flow was sub-sampled with a 4 m length of ¼-inch
PFA tubing that extracted the required 3300 sccm air from the centre of the stainless steel pipe and directed it through a 1
µm PFA filter to the $NO_3$-reactivity instrument. Due to thermostat break-down during NOTOMO, the $NO_3$-reactivity
measurements were performed with the flow-tube at container temperature, which was variable (14 - 31 °C).
Previous campaigns at the Taunus Observatory have revealed occasionally high night-time mixing ratios of $NO_3$ and $N_2O_5$
(Sobanski et al., 2016b). As sampling $NO_3$ and $N_2O_5$ from ambient air would bias the $NO_3$-reactivity measurements to low
values, a 2 l glass flask heated to $\approx$ 40-50 °C was placed at night in the ambient air stream to decompose $N_2O_5$ to $NO_3$ and
$NO_2$. Based on its thermal dissociation rate coefficient (0.75 $s^{-1}$ at 50 °C), $N_2O_5$ completely decomposes within the $\approx$ 40 s
residence time in this glass vessel, and the $NO_3$ formed is expected to be lost on the uncoated glass walls, thus preventing
reformation of $N_2O_5$. Measurements with $\approx$ 200 pptv of $N_2O_5$ added directly to the heated vessel and measured by the
ambient and heated channels of the two-cavity CRDS (see section 2.2) confirmed that neither $NO_3$ nor $N_2O_5$ survived. As the
$N_2O_5$ mixing ratio was measured during NOTOMO it is in principal possible to correct the data for the additional $NO_2$ thus
generated. However, on most nights $N_2O_5$ levels were too low for this to have a significant effect. Further experiments with





isoprene and α-pinene indicated that there was no significant change in $NO_3$-reactivity when the glass vessel was used or not,
indicating no significant losses of these VOCs in the glass flask. We cannot exclude that other, less volatile organic trace
gases including e.g. acids or peroxides may be lost in the glass vessel, but these are not expected to contribute significantly
to $NO_3$ losses as their rate coefficients for reaction with $NO_3$ are generally too low. A further potential bias related to the use
of the glass trap is the thermal decomposition of PAN and related peroxy nitrates, which can acquire concentrations of up to
a few ppb at this site (Thieser et al., 2016; Sobanski et al., 2016c). If PAN decomposes in the glass vessel $NO_2$ will form,
thus contributing to the measured reactivity. Simulations indicate that during the 40 s residence time in the heated flask (at
50 °C) only a small fraction ($\approx 2.6$ %) of the PAN decomposes to form $NO_2$. For future experiments in environments of high
$NO_x$ with $N_2O_5$ and $NO_3$ present, the system will be operated at a lower temperature (e.g. 35 °C, $\tau_{PAN}$ = ~500 s, $\tau_{N2O5}$ = ~6 s)
to make sure all of the $N_2O_5$/$NO_3$ is removed but PAN is preserved. We note that when measuring $NO_3$-reactivity in regions
with large biogenic emissions, the use of the glass vessel to remove $NO_3$ and $N_2O_5$ is generally not necessary as high levels
of biogenic VOCs and the low levels of $NO_x$ often found in forested / rural environments remote from anthropogenic
influence will result in very low levels of $NO_3$ or $N_2O_5$.
During NOTOMO, ambient levels of $NO_2$, $NO_3$, $N_2O_5$ and organic nitrates were measured with the CRDS instruments
previously described by Sobanski et al. (Sobanski et al., 2016a; Thieser et al., 2016). The uncertainty in the measurements
was 8 % for $NO_2$, 20 % for $NO_3$ whereas the uncertainty for PAN was highly variable for each data point (Sobanski et al.,
2016c). The $O_3$ mixing ratios were measured using a dual beam ozone monitor (2B-Technology Model 202) with an
uncertainty of 2 %. [NO] was not directly measured but its day-time concentration was calculated assuming photo-
stationary-state via expression (12):
$[NO]_{calc} = J(NO_2) [NO_2] / k_{(NO+O3)}[O_3]$             (12)
where $J(NO_2)$ is the photolysis frequency of $NO_2$ and $k_{(NO+O3)}$ is the rate constant for reaction of NO with $O_3$. This expression
ignores the oxidation of NO to $NO_2$ via e.g. reactions of peroxy radicals and thus overestimates NO. $J(NO_2)$ was measured
using a spectral radiometer located close to the inlet (MetCon).
In this manuscript we focus on a three-day period, during which $NO_3$-reactivity was measured (Fig. 12a). The $NO_3$
reactivity, $k_{RTG}$, varied from 0.005 to 0.1 $s^{-1}$ during night-time but reached values as high as 1.4 $s^{-1}$ during day-time. The total
uncertainty of the measurement is depicted by the green, shaded area. The red line indicates that, as expected, day-time
losses are dominated by reaction with NO (up to 1.3 $s^{-1}$). Night-time values of $k_{RTG}$ were between 0.005 and 0.1 $s^{-1}$.
Assuming that NO levels are close to zero as measured previously at this site during night-time (Crowley et al., 2010), $k_{RTG}$
is then expected to be dominated by VOCs.
In Fig. 12b, we compare values of $k_{RTG}$ obtained by rigorous data correction (black curve), to those calculated directly from
expression (11) (blue curve). The simple analytical expression (blue line) results in an underestimation of the reactivity,
especially during night, when the overall reactivity is low, and in periods of high [$NO_2$]. Owing to lack of temperature
stabilization of the darkened reactor (at this time not yet incorporated) and break-down of the flow-tube thermostat during
the campaign, temperature fluctuations in the container resulted in $MDC_{NO3}$ = 5.6 pptv and hence an average, measureable





reactivity of $\approx 0.01$ s$^{-1}$ during the campaign. As described in section 5 the minimum detectable change in NO$_3$ was combined
with the uncertainty associated with the dilution factor, reaction time, [NO$_2$], [PAN] and rate constants used to calculate the
overall uncertainty for the reactivity at every data point. The overall uncertainty for the measurement period illustrated in
Fig. 12 was $\approx 25$ %.
In Fig. 13a/b we compare the measured night-time NO$_3$-reactivity with that obtained from the stationary-state analysis using
expression (1). For the two nights in the period analysed, NO$_3$ mixing ratios were between 5 and 37 pptv ([NO$_3$]>> 5 pptv)
and the calculated stationary-state loss rate coefficients varied between 0.03-0.003 s$^{-1}$ compared to the measured reactivity
which was between 0.05-0.006 s$^{-1}$ with a short time period in which $k_{RTG}$ fell below the detection limit of the instrument.
Within the total uncertainty, the measured and stationary-state reactivities are in reasonable agreement for most of the night
from the 17$^{th}$ to the 18$^{th}$. From the night 18$^{th}$ to the 19$^{th}$ the stationary-state reactivity is much lower (up to a factor of eight)
than that measured. This difference and also the higher variability can be attributed to rapid variations in concentrations of
VOCs at the inlet (due e.g. to emissions from nearby trees) that are not considered in the stationary-state approach; i.e. very
local emissions of reactive gases will result in breakdown of the stationary-state assumption leading to the underestimation
of the reactivity of the local mixture of VOCs and NO$_x$. As the direct measurement of the NO$_3$ reactivity with this device
sums over all VOCs present in the air mass sampled, it should give the same result as summing each VOC concentration
multiplied by the individual rate coefficients for reaction with NO$_3$, i.e NO$_3$ reactivity = $\sigma$ [VOC]$_i k_i$. As demonstrated
previously for this mountain site (Sobanski et al., 2016b), summed losses based on measurement of VOCs can significantly
exceed the reactivity based on a stationary-state analysis especially under some meteorological situations in which a low-
lying residual layer (with high NO$_3$ concentrations) influences the measurement.

## 7 Conclusion and outlook

We present the first instrument for measurement of NO$_3$ reactivity in ambient air. The flow-tube based instrument, utilizes
the depletion of synthetically generated NO$_3$ when mixed with ambient air and has a dynamic range of 0.005 s$^{-1}$ to 45s$^{-1}$.
Following intensive laboratory characterization to determine the effective reaction time, the wall loss constant of NO$_3$ and
the effect of NO$_3$ formation and reformation in the flow-tube, it was successfully tested against an isoprene standard. The
overall uncertainty depends on the relative rate of reaction of NO$_3$ with NO$_2$ or with other traces gases (e.g. VOCs or NO)
that do not generate N$_2$O$_5$ and which, under ideal conditions, is close to 15 %. The instrument is thus best suited for
measurement of NO$_3$ reactivity in regions with high biogenic activity and relatively low direct anthropogenic emissions of
NO$_x$, i.e. regions where the measurement of NO$_3$ concentrations is difficult owing to low production rates and a high loss
term.
First deployment of the instrument was during the NOTOMO observational experiment in summer 2015 at a forested,
mountain site with urban influence. The measured NO$_3$ reactivity ranged from 0.006 to 0.1 to s$^{-1}$ at night-time and reached
values as high as 1.4 s$^{-1}$ during day-time. As expected, day-time reactivity was dominated by reaction with NO while night-





time reactivity involved other (presumably organic) trace gases. A comparison with stationary-state calculations of the $NO_3$
reactivity revealed poor agreement on occassions, presumably related to very local emissions causing a breakdown of the
stationary-state assumption.
Improvements to the dynamic range of the instrument require further stabilization of the $NO_3$ source and cavity-optics to
reduce the minimal detectable change in $NO_3$ (presently $MDC_{NO3} = 2.5$ pptv). This could also be achieved by the use of
larger volume flow-tubes. Future deployment with simultaneous measurements of $NO_3$, $NO_2$, $O_3$ and VOCs will be
conducted to compare direct measurements of $NO_3$ reactvity with those obtained from the stationary-state approach and also
those calculated from summing losses to individual VOCs.
**Acknowledgements**
We would like to thank Heinz Bingemer and the staff and department of the Johann Wolfgang Goethe–University, Frankfurt
am Main for logistical support and access to the Taunus Observatory during NOTOMO. We also would like to thank Eva
Pfannerstil for providing the isoprene standard. We thank DuPont for the sample of FEP used to coat the walls of the flow
tube and darkened reactor. This work was carried out in partial fulfilment of the PhD (Johannes Gutenberg University,
Mainz, Germany) of Jonathan Liebmann.





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



**Table 1:** Facsimile[1] Simulations

| | | |
|---|---|---|
| $NO_2 + O_3 \rightarrow NO_3 + O_2$ | $k_1 = 3.52 \times 10^{-17}$ cm$^3$ molecule$^{-1}$ s$^{-1}$ | $k_1$ |
| $NO_3 + NO \rightarrow 2\ NO_2$ | $k_2 = 2.60 \times 10^{-11}$ cm$^3$ molecule$^{-1}$ s$^{-1}$ | $k_2$ |
| $NO_3 + NO_2 + M \rightarrow N_2O_5 + M$ | $k_3 = 1.24 \times 10^{-12}$ cm$^3$ molecule$^{-1}$ s$^{-1}$ | $k_3$ |
| $N_2O_5 + M \rightarrow NO_2 + NO_3 + M$ | $k_4 = 4.44 \times 10^{-2}$ cm$^3$ molecule$^{-1}$ s$^{-1}$ | $k_4$ |
| $NO + O_3 \rightarrow NO_2 + O_2$ | $k_5 = 1.89 \times 10^{-14}$ cm$^3$ molecule$^{-1}$ s$^{-1}$ | $k_5$ |
| $NO_3 + wall \rightarrow NO_2$ | $k_w = 4 \times 10^{-3}$ s$^{-1}$ | $k_w$ |
| $k_{RTG}$ | variable / fitted | $k_{RTG}$ |

[1]For all simulations FACSIMILE-CHEKMAT (Release H010 DATE 28.04.87 Version 1) was used. The rate constants ($k_i$) listed were
taken from the IUPAC recommendations (Atkinson et al., 2004; IUPAC, 2016) at 298 K and 1 bar.

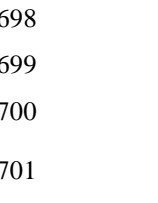






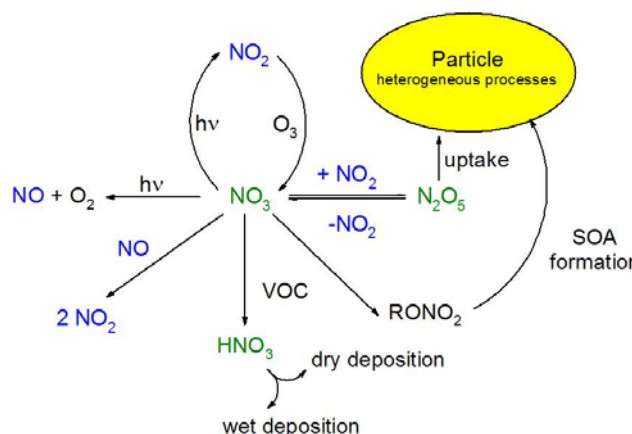

**Figure 1:** Gas-phase formation and loss of tropospheric $NO_3$.

SOA = secondary organic aerosol, $RONO_2$ are alkyl-nitrates.



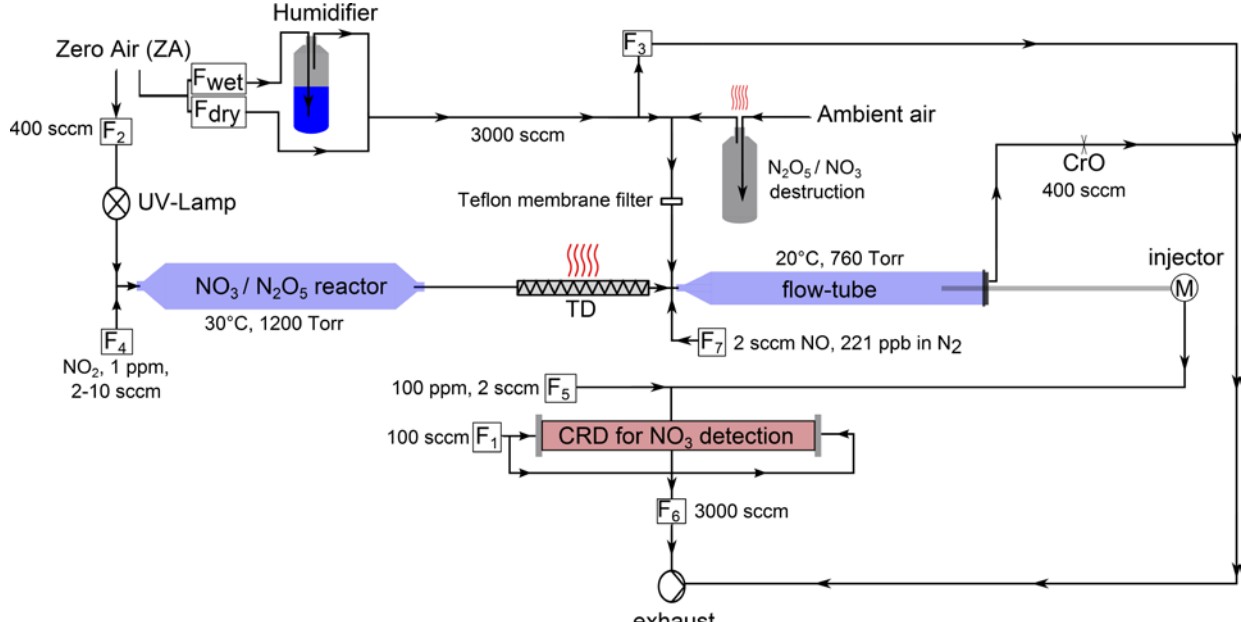

**Figure 2:** Schematic diagram of the NO$_3$-reactivity measurement. F$_1$-F$_7$ are mass flow-controllers: F$_1$ = mirror purge flow, F$_2$ = zero-air for O$_3$ generation, F$_3$ = dilution / inlet overflow (switching between zero-air and ambient), F$_4$ = NO$_2$ for NO$_3$ / N$_2$O$_5$ generation, F$_5$ = NO titration of NO$_3$, F$_6$ = cavity flow to pump, F$_7$ = NO flow for online reactivity calibration. CrO = critical orifice. TD = heated tubing for thermal decomposition of N$_2$O$_5$ to NO$_3$ at 140 °C.















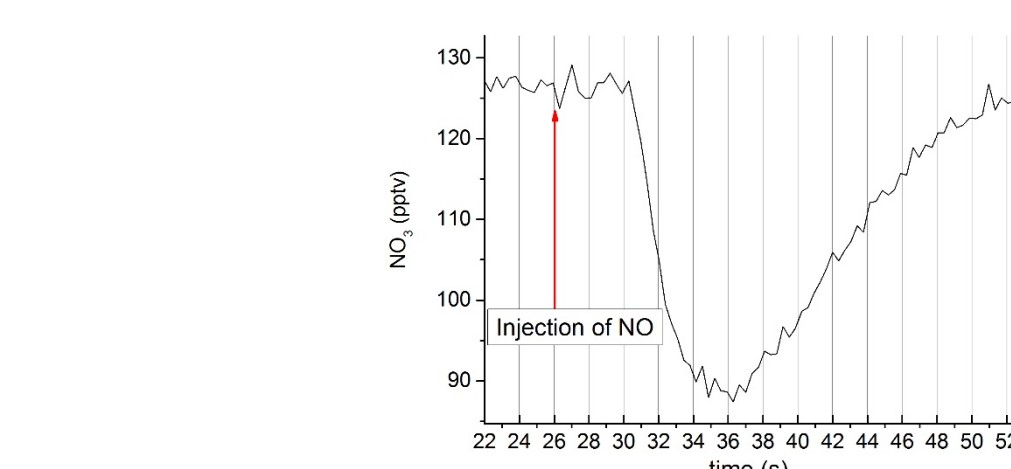




**Figure 3:** Derivation of effective reaction time by addition of a pulse
(at t = 26 s) of NO using a syringe. The subsequent depletion in the
NO$_3$ signal was analysed using expression 4.

















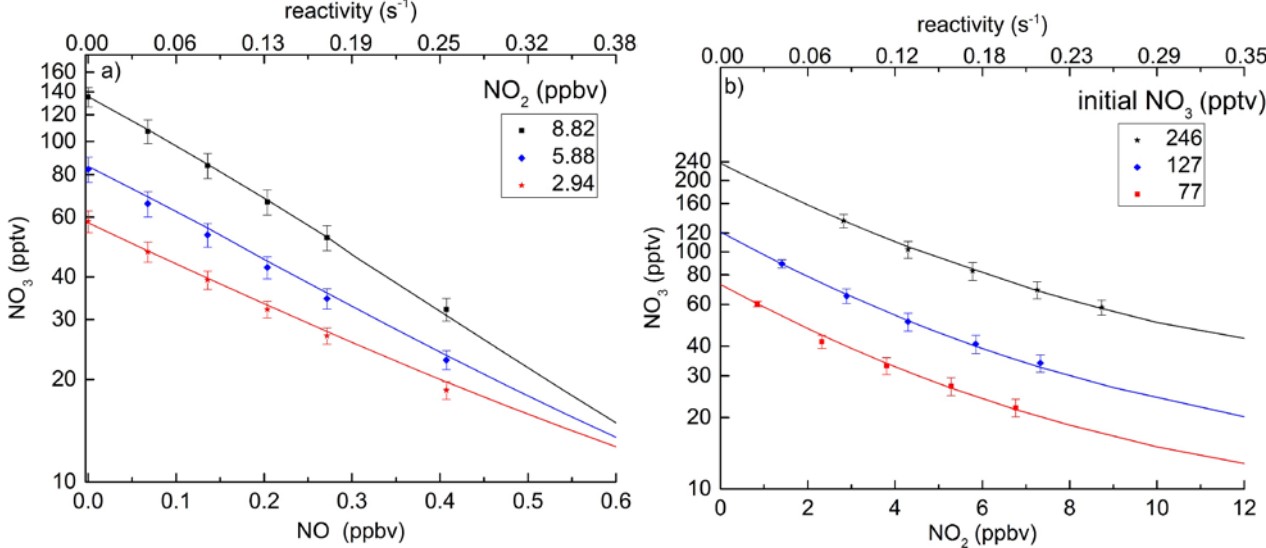

**Figure 4:** Characterisation of the flow-tube by numerical simulation of the $NO_3$ change following addition of NO and $NO_2$ at different mixing ratios. The symbols are measured $NO_3$ mixing ratios, the lines are the results of numerical simulations. The reactivity scales were calculated from $k_2[NO]$ and $k_3[NO_2]$ using the rate constants listed in Table 1.



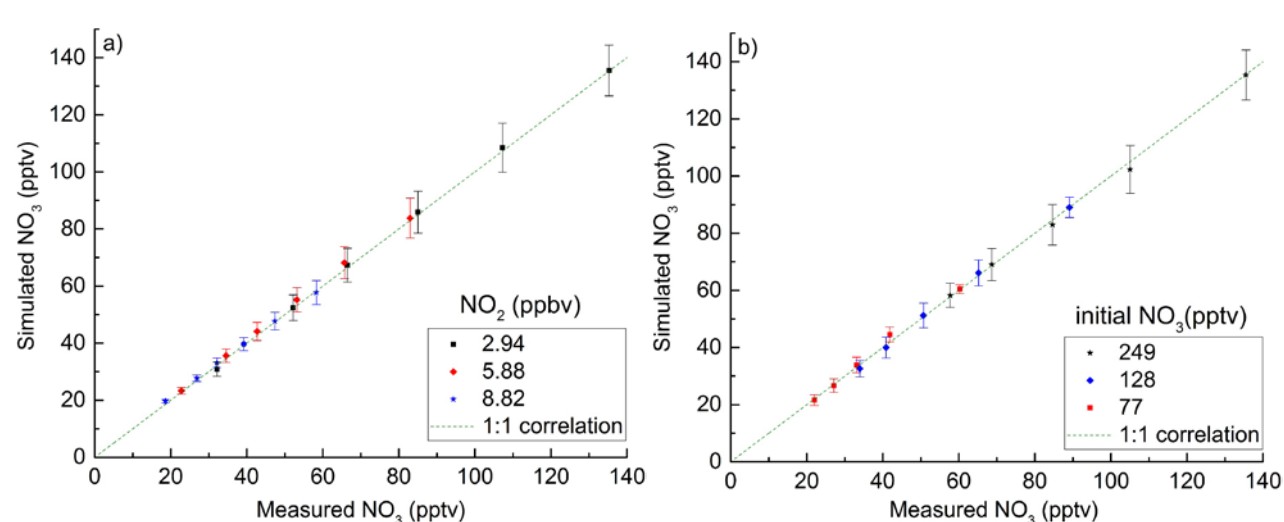

**Figure 5:** *Left:* Measured versus simulated [NO$_3$] for different amounts of added NO (67, 134, 201, 268, 402 pptv) and at three different mixing ratios of NO$_2$. *Right*: Measured versus simulated NO$_3$ (initially 77, 128 or 249 pptv) at different amounts (1.5, 3, 4.5, 6 ppbv) of added [NO$_2$]. The solid lines represent 1:1 agreement.





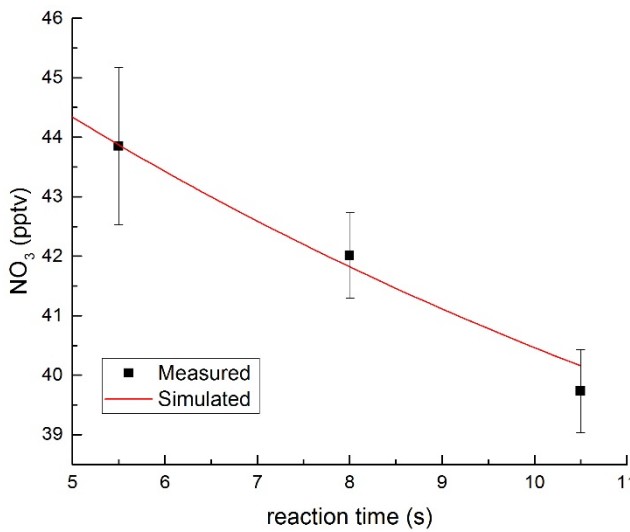

**Figure 6:** Determination of the wall loss constant of $NO_3$ by variation of the reaction time (injector position). The simulation indicates a wall loss constant of $k_w = 0.004$ s$^{-1}$.




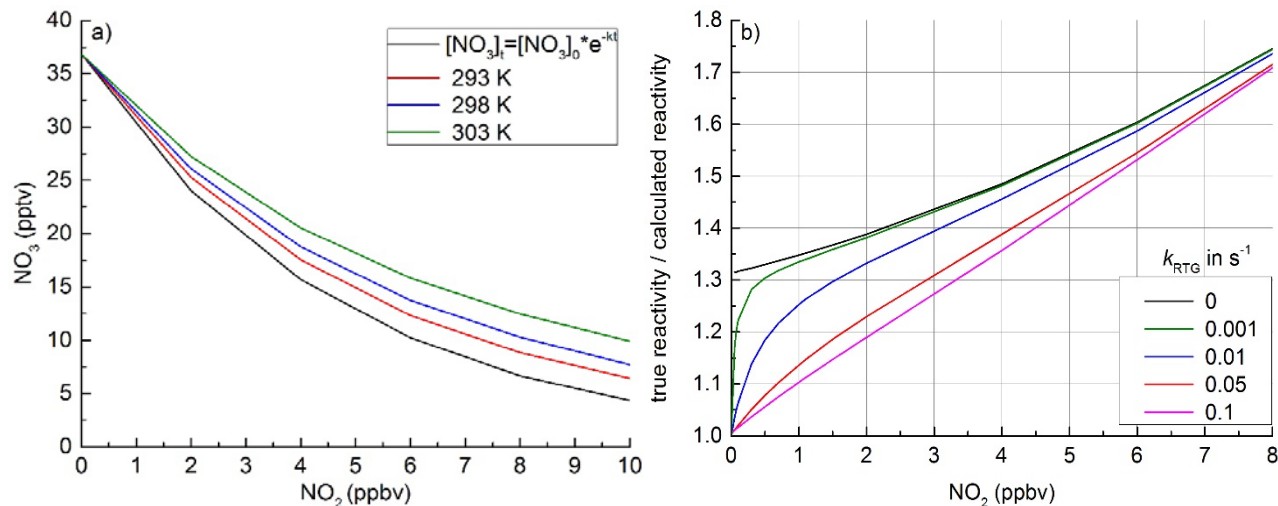

**Figure 7:** Influence of $N_2O_5$ formation and decomposition in the flow-tube. a) simulated (red, blue and green) mixing ratio of $NO_3$
versus added $NO_2$ at a reaction time of 10.5 s at various temperatures and thus thermal decomposition rates of $N_2O_5$. The simple
exponential decay of $NO_3$ (expression 9) is given by the black line. b) Effect of $NO_2$ level on the ratio of true reactivity / reactivity
calculated from expression (8) for different loss rate constants for $NO_3$ reacting with reactive traces gases.














**Figure 8:** Simulated NO$_3$ production in the flow-tube at different O$_3$ and

NO$_2$ mixing ratios at a fixed reaction time of 10.5 s.























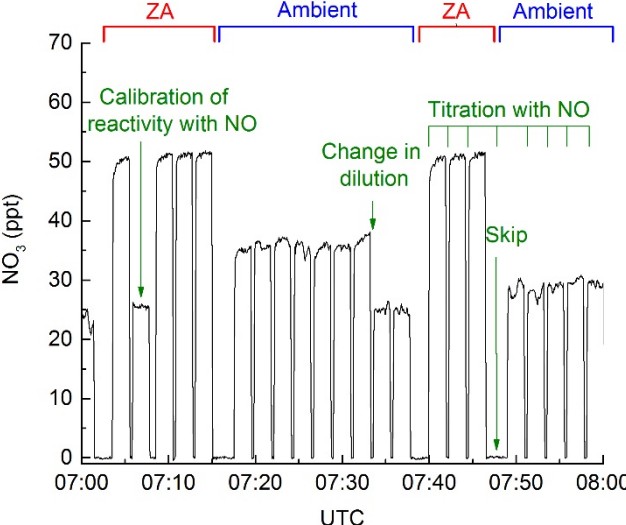

**Figure 9:** Raw data showing the change in $NO_3$ (10.5 s reaction time) between zero-air (ZA, periods marked with red brackets) and ambient air (Ambient, blue brackets). The Figure also shows periods of titration of $NO_3$ with NO (≈ 2 min intervals, green brackets), a change in the dilution factor from 4 to 3 (at ≈ 07:33) and an in-situ reactivity calibration (at ≈ 07:07). The "skip" periods are those in which data is not analysed due to switching from ambient air to zero-air and vice versa.























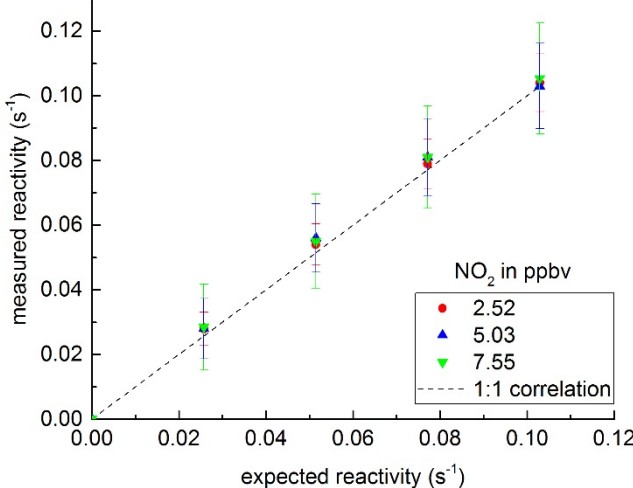

**Figure 10:** Verification of the experimental procedure by addition of isoprene at different $NO_2$ mixing ratios. The known reactivity was calculated from the isoprene mixing ratio (1.5 – 6 ppbv) and the rate coefficient for reaction of isoprene with $NO_3$. Experiments were performed in dry zero-air. The error bars in the simulation are due to uncertainties in [isoprene] and [$NO_2$] (both 5 %) and the reaction time (10 %).































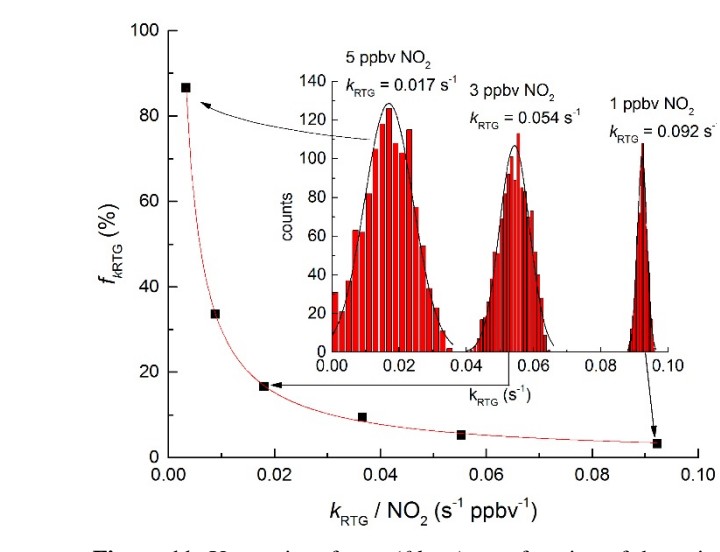

**Figure 11:** Uncertainty factor ($f\,k_{RTG}$) as a function of the ratio $k_{RTG}$ / [NO$_2$] as derived from Monte-Carlo simulations. The relationship (red curve) is described by $f\,(k_{RTG}) = 0.33 \times (k_{RTG} /$ NO$_2$)-0.977. The results of three individual sets of 1200 simulations are shown as histograms.





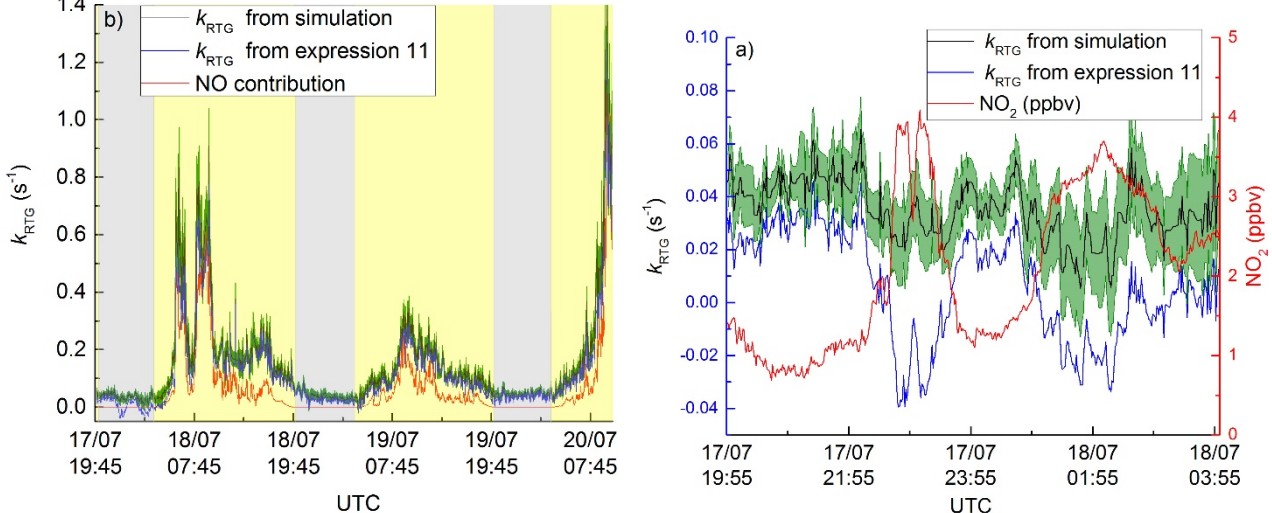

**Figure 12:** a) Measured values of $k_{RTG}$ over a 3 day period. The overall uncertainty is represented by the green, shaded area. The black lines are $k_{RTG}$ obtained by full simulations, the blue lines are calculated using expression (11) (without correction for $N_2O_5$ formation and decomposition). The contribution of NO to the $NO_3$ reactivity is displayed as the red line. Yellow regions correspond to day-time, grey regions correspond to night-time b) Zoom in on a night-time period with low reactivity emphasizing the effect of $NO_2$-induced formation and decomposition of $N_2O_5$.





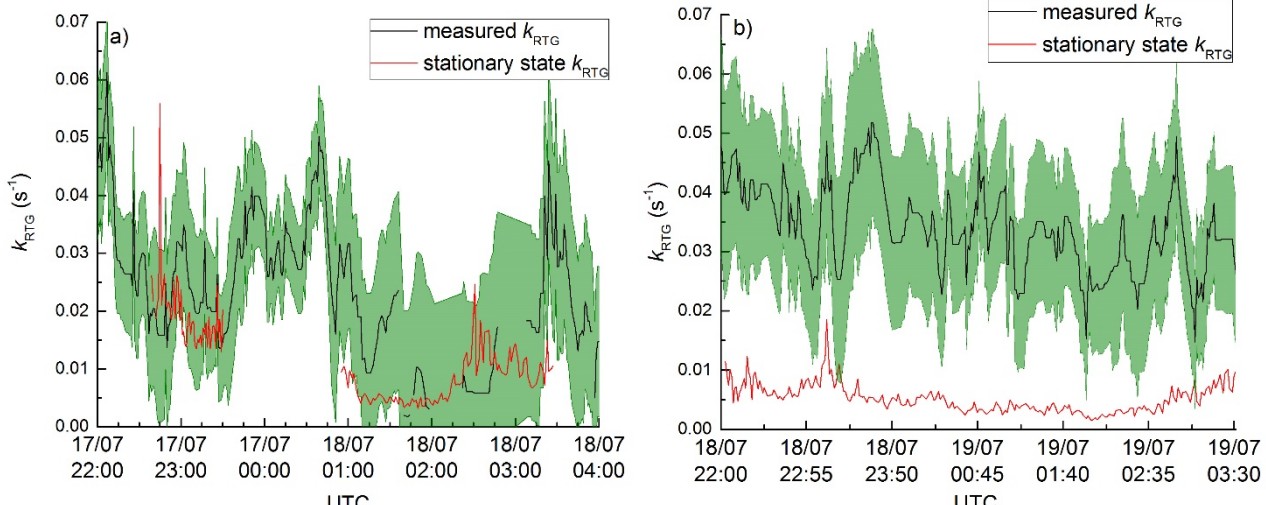

**Figure 13:** Comparison of stationary-state and measured $NO_3$ loss rates. Uncertainty in $k_{RTG}$ (see text) are displayed as the green shaded areas.