# Peer review of "Measurement of ambient NO3 reactivity: Design, characterization"

_Atmospheric Measurement Techniques, 2016_

## Referee Comment (RC2) · S. Brown (Referee) · 5 Feb 2017

General Comments

This paper describes the first demonstration of NO3 radical reactivity and its use in ambient air. Reactivity measurements for HOx radicals have become a routine feature of instruments that measure ambient HOx concentrations, yet no analogous instrument has been developed for NO3 until now. As such, the paper represents an important contribution that will be of interest to the readership of AMT and that may serve as a seminal work that stimulates a new generation of atmospheric instrumentation. With one exception (see below), the development is thorough and convincing. It should be published in AMT.

[Figure]

The major general comment is the lack of discussion of secondary chemistry in the NO3 flow tube. While the authors thoroughly consider the effect of the NO3-N2O5 equilibrium and the potential production of NO3 from NO2 + O3 in the flow tube, they do not comment (unless I missed this point) on the potential perturbation to the NO3 reactivity by the addition of NO3 itself to the ambient air in the flow tube. For example, at the lower end of the measurement range, k = 0.005 s 1, only about 30 pptv of a typical monoterpene ($\alpha$-pinene) would be required to produce this reactivity, smaller than the added NO3. The procedure calls for addition of 50 pptv NO3 at that start of the flow tube (and in some experiments, more was used). The NO3 reaction would lead to ∼8% reduction of the initial $\alpha$-pinene mixing ratio during the 10.5 s flow tube residence time (taking kinetics as first order for simplicity, even though in this example they are actually second order), placing some bias on the derived reactivity. Furthermore, the rapid VOC degradation induced by NO3 addition would presumably lead to formation of peroxy radicals and / or HOx that could further react with NO3 or the ambient VOC levels. If reactivity were dominated by even more reactive VOCs (e.g., sesquiterpenes), these effects could become significant. The authors should consider the potential for such secondary chemistry to bias the NO3 reactivity measurement under the chosen level of added NO3.

Aside from this comment, the authors should consider the following more specific comments prior to publication.

Specific Comments

Line 32: should add the phrase "at elevated NOx", since in the absence of NOx, O3 is the major nighttime oxidizing agent for these compounds.

Line 66: Equation appears to be inverted. Units as written would be s, not s 1, so this should be Tau_ss, not 1/Tau_ss

Lines 99-100: Both Fuchs et al. (Anal. Chem. 80, 6010, 2008) and Wagner et al. (AMT, 4, 1227, 2011) describe the use of crystalline N2O5 for calibration of NO3 and

N2O5 instruments in the field in which there is no significant NO2 impurity. In-field synthesis and sample stability have also been achieved, albeit with some difficulty. While this reviewer agrees that the method is difficult and the in-situ generation source may be superior, it may be worth pointing out that it is not impossible. If achievable, the crystalline source would have the advantage of no O3 and reduced NO2.

Lines 104-105: Is the 0.93 ppbv mixing ratio of NO2 before or after its dilution into the O2 flow? Presumably after, but not clear from the way it is written.

Lines 108-111: The termolecular reaction of NO2+NO3 is not rate limiting in this system, so it is difficult to see (intuitively) why high pressure would make N2O5 production more efficient, or why the reactor would be sensitive to fluctuations in external pressure.

Line 114: Reaction does not go to completion in 5 minutes at 400 ppbv O3, correct?

Line 122: NO2 (1 ppbv) lower after the reactor than before? (see comment above) Must be an error in the numbers given above.

Line 229: "non-isothermal effects" is not clear

Line 252-262: The role of the N2O5 equilibrium and the excess O3 in the flow tube should be obvious enough that explicit discussion of equation (6) and deviations from it are not really needed. Suggest omitting this simple expression in favor of the discussion of the more accurate numerical simulations to simplify the paper.

Lines 291-292: Is there any degradation of kw observed during field sampling in ambient air? If NO3 has a large wall loss rate constant on glass, it would presumably also be lost readily to FEP coatings there were not pristine.

Line 297: Does expression (7) take radial diffusion limitation into account? Also, r and c are presumably the tube radius and mean molecular speed of NO3, but should be defined in the text.

Line 360-362: Statement could be stronger than "we prefer". The numerical simulation

is obviously more accurate and more general, and can be described as such.

Line 371: Is this really the time for flushing reactive gases? Several minutes for a system with 10.5 s residence time?

Line369-382: The authors may elect to shorten this section, which could be conveyed in a sentence or two to state that an iterative fit was used.

Line 410: Is the 0.2 pptv 1-sigma?

Line 514, Figure 12a: Figure would be much more effective with the y-axis on a log scale to illustrate the difference between night and day. It is difficult to see the nighttime reactivity on this scale, and the quantitative day / night contrast should be of interest to the readers.

―――――――――――――――――――

---

## Author Comment (AC1) · 1 Mar 2017

The comment was uploaded in the form of a supplement:
http://www.atmos-meas-tech-discuss.net/amt-2016-381/amt-2016-381-AC1-supplement.pdf

---

## Author Comment (AC2) · 1 Mar 2017

**Referee 2 (Steven Brown)**

In the following, the referee's comments are reproduced (black) along with our replies (blue) and changes made to the text (red) in the revised manuscript.

**General statement:**

General Comments This paper describes the first demonstration of NO3 radical reactivity and its use in ambient air. Reactivity measurements for HOx radicals have become a routine feature of instruments that measure ambient HOx concentrations, yet no analogous instrument has been developed for $NO_3$ until now. As such, the paper represents an important contribution that will be of interest to the readership of AMT and that may serve as a seminal work that stimulates a new generation of atmospheric instrumentation. With one exception (see below), the development is thorough and convincing. It should be published in AMT.

We thank Steven Brown for this thorough review and overall positive assessment of our manuscript. The manuscript has been improved in line with his comments listed below.

The major general comment is the lack of discussion of secondary chemistry in the NO3 flow tube. While the authors thoroughly consider the effect of the NO3-N2O5 equilibrium and the potential production of NO3 from NO2 + O3 in the flow tube, they do not comment (unless I missed this point) on the potential perturbation to the NO3 reactivity by the addition of NO3 itself to the ambient air in the flow tube. For example, at the lower end of the measurement range, k = 0.005 s 1, only about 30 pptv of a typical monoterpene (α-pinene) would be required to produce this reactivity, smaller than the added NO3. The procedure calls for addition of 50 pptv NO3 at that start of the flow tube (and in some experiments, more was used). The NO3 reaction would lead to ~8% reduction of the initial α-pinene mixing ratio during the 10.5 s flow tube residence time (taking kinetics as first order for simplicity, even though in this example they are actually second order), placing some bias on the derived reactivity.
Furthermore, the rapid VOC degradation induced by NO3 addition would presumably lead to formation of peroxy radicals and / or HOx that could further react with NO3 or the ambient VOC levels. If reactivity were dominated by even more reactive VOCs (e.g., sesquiterpenes), these effects could become significant. The authors should consider the potential for such secondary chemistry to bias the NO3 reactivity measurement under the chosen level of added NO3.

We now address this briefly in section 2.1: "We later assess the potential change in air-mass reactivity (i.e. by depletion of reactive trace gases of formation of reactive radicals) following addition of $NO_3$ at these levels to ambient air."

And in detail in section 5 in which we write: "We now examine the potential bias caused by use of $NO_3$ concentrations as large as 50 pptv, which may change the reactivity of the air either by removing a significant fraction of gas-phase reactants or via formation of peroxy radicals ($RO_2$), which may also react with $NO_3$. In a first scenario, we assume that the reactivity is caused by a single species, namely the generally dominant terpene, α-pinene and consider both low ($k_{RTG} = 0.005$ s$^{-1}$) and high reactivity regimes ($k_{RTG} = 0.1$ s$^{-1}$). A value of $k_{RTG} = 0.005$ s$^{-1}$ would result if 34 pptv of α-pinene were available for reaction. In a first approximation, assuming first-order kinetics, we calculate that 2.5 pptv of the initially available 50 pptv of $NO_3$ are lost in the 10.5 s

reaction time, and consequently a change in α-pinene of 2.5 pptv would also occur. This is only 7 % of the initial concentration, indicating an upper limit to a negative bias of 7 %. This is an upper limit as the assumption of first order kinetics is not entirely appropriate. As $NO_3$ reacts with α-pinene in air to form a nitroxy peroxy radical ($RO_2$) we also consider a positive bias due to reaction of $NO_3$ with this $RO_2$. To do this we assume a rate constant of $1.2 \times 10^{-12}$ $cm^3$ molecule$^{-1}$ s$^{-1}$ for the reaction as observed for $NO_3 + CH_3O_2$ (Atkinson et al., 2006) and assume that this rate constant is independent of the nature of the organic fragment (R) as is the case for reactions of $RO_2$ with NO. The 2.5 pptv $RO_2$ thus generated results in an incremental $NO_3$ reactivity of $7 \times 10^{-5}$ s$^{-1}$, a positive bias of 1.5 %. Again, this is an upper limit, as the calculation assumes that this concentration of $RO_2$ is constant and available for the whole 10.5 s of reaction time. For higher reactivity (0.1 s$^{-1}$) a similar calculation shows that the 670 pptv required would reduce the $NO_3$ concentration to 17.5 pptv, itself being diminished to ~640 pptv, a change of just 5 %. The 32.5 pptv $RO_2$ generated would result in a loss rate constant for $NO_3$ of ~$9 \times 10^{-4}$ s$^{-1}$, a positive bias of ~ 1%. In conclusion, for reactive systems in which a large concentration of reactive trace gases with moderate reactivity towards $NO_3$ are encountered, we expect no significant bias. The only scenario, in which a large bias can ensue, is when a low reactivity is caused by a very low concentration of an extremely reactive traces gas. Taking the example of 1 pptv of a highly reactive terpenoid ($k = 2 \times 10^{-10}$ $cm^3$ molecule$^{-1}$ s$^{-1}$) it is easy to show that it would be reduced to just a few percent of its initial concentration when mixed with 50 pptv of $NO_3$ for 10.5 s. In this case a large negative bias would result. In the real atmosphere, this situation is however unlikely to occur as such reactive species are usually substantially reduced in concentration compared to the generally dominant biogenics such as α-pinene."

And in the conclusions where we write: "Reduction in the initial $NO_3$ concentration used would also reduce any potential bias caused by depletion of reactants or secondary chemistry."

Aside from this comment, the authors should consider the following more specific comments prior to publication.

Line 32: should add the phrase "at elevated NOx", since in the absence of NOx, O3 is the major nighttime oxidizing agent for these compounds.
We have added the phrase to the text: "As OH levels are vastly reduced in the absence of sunlight, the $NO_3$ radical (formed by reaction of $NO_2$ with $O_3$, R1) is the major oxidizing agent at elevated $NO_x$ for many biogenic terpenoids and other unsaturated compounds at night-time….."

Line 66: Equation appears to be inverted. Units as written would be s, not s-1, so this should be Tau_ss, not 1/Tau_ss. Corrected

Lines 99-100: Both Fuchs et al. (Anal. Chem. 80, 6010, 2008) and Wagner et al. (AMT, 4, 1227, 2011) describe the use of crystalline N2O5 for calibration of NO3 and C2 N2O5 instruments in the field in which there is no significant NO2 impurity. In-field synthesis and sample stability have also been achieved, albeit with some difficulty. While this reviewer agrees that the method is difficult and the in-situ generation source may be superior, it may be worth pointing out that it is not impossible. If achievable, the crystalline source would have the advantage of no O3 and reduced NO2.
We have amended the text accordingly: "The generation of $NO_3$ from gas-phase $N_2O_5$ eluted from samples of crystalline $N_2O_5$ (at -80 °C) was found to be insufficiently stable for the present

application and is also difficult (though not impossible, see e.g. (Fuchs et al., 2008; Wagner et al., 2011)) to use during field campaigns where adequate laboratory facilities for the safe generation and purification of $N_2O_5$ are frequently not available. In addition, generation of $NO_3$ from $N_2O_5$ was also accompanied by an $NO_2$ impurity of several parts per billion (ppbv).

Lines 104-105: Is the 0.93 ppbv mixing ratio of NO2 before or after its dilution into the O2 flow? Presumably after, but not clear from the way it is written.
There was an error in the units, it should have been 0.93 ppmv. This has been corrected.

Lines 108-111: The termolecular reaction of NO2+NO3 is not rate limiting in this system, so it is difficult to see (intuitively) why high pressure would make N2O5 production more efficient, or why the reactor would be sensitive to fluctuations in external pressure.
The termolecular reaction with $NO_2$ competes with loss of $NO_3$ to the walls. Increasing the pressure increases the fraction of $NO_3$ that forms $N_2O_5$. Higher pressures compared to ambient ($\approx$ factor 2) help to decouple from pressure fluctuations as the flow rate through the orifice coupling the two vessels tends towards critical operation.

Line 114: Reaction does not go to completion in 5 minutes at 400 ppbv O3, correct?
Correct. We add text to explain that only a small fraction of the $NO_2$ is initially converted to $NO_3$.: "The approximate reaction time for the stepwise conversion of $NO_2$ to $N_2O_5$ in the darkened reactor is $\approx$ 5 min. Based on the $O_3$ concentration and the rate constant for R1, the initial conversion of $NO_2$ to $NO_3$ is about 15%."

Line 122: $NO_2$ (1 ppbv) lower after the reactor than before? (see comment above) Must be an error in the numbers given above. Yes, the units of the $NO_2$ mixing ratio were wrong. See answer above (lines 104-105).

Line 229: "non-isothermal effects" is not clear.
This refers to temperature inhomogeneities in the flow-tube and is described in detail by Huang et al.

Line 252-262: The role of the N2O5 equilibrium and the excess O3 in the flow tube should be obvious enough that explicit discussion of equation (6) and deviations from it are not really needed. Suggest omitting this simple expression in favor of the discussion of the more accurate numerical simulations to simplify the paper.
The deviation from ideal behaviour is often a clue to the quality of a more complicated numerical model. We show that deviation if small when $NOx$ levels are low indicating that any errors in the numerical procedure are likely not to result in large systematic errors. We feel this makes the paper more readable and prefer to keep the "ideal case" in.

Lines 291-292: Is there any degradation of kw observed during field sampling in ambient air? If $NO_3$ has a large wall loss rate constant on glass, it would presumably also be lost readily to FEP coatings there were not pristine.
$k_w$ was not measured by injector movement during field campaign. However, there was no indication of enhanced wall loss rate constants after the campaign compared to before indicating no significant degradation of the FEP coating. "We write: Similar experiments performed before and after the NOTOMO campaign (see below) indicated that the FEP coating did not degrade

significantly following sampling of filtered, ambient air." In order to avoid ambiguity, we have removed the sentence describing loss of $NO_3$ on the uncoated wall.

Line 297: Does expression (7) take radial diffusion limitation into account? Also, r and c are presumably the tube radius and mean molecular speed of NO3, but should be defined in the text. We have changed the text to clarify: "Using expression (7) where r is the flow-tube radius, $\bar{c}$ the mean molecular speed and which assumes laminar flow and no diffusive limitation to uptake, this value of $k_w$ can be converted to an approximate uptake coefficient for $NO_3$ to the FEP-coated tube of $\approx 5x10^{-7}$."

Line 360-362: Statement could be stronger than "we prefer". The numerical simulation C3 is obviously more accurate and more general, and can be described as such. Agreed. We now simply state that we use the numerical simulations.

Line 371: Is this really the time for flushing reactive gases? Several minutes for a system with 10.5 s residence time?
Normally, ~3 times the residence time (~32 s) are commonly used to flush 95 % of gases from a cylindrical flow tube. This is also apparent from the $NO_3$ behaviour when adding NO via the syringe (section 2.3.1). This time can be extended if the geometry is not ideal (e.g. presence of dead volumes in fittings) and if the gases have finite residence times on the inlet surfaces (i.e. are sticky). We now write: The data show that a plateau in the $NO_3$ signal with zero-air is observed after about 2-3 titration cycles are complete, which is the result of slow flushing through the inlet of reactive gases, which have extended surface residence times on the inlet material and fittings.

Line369-382: The authors may elect to shorten this section, which could be conveyed in a sentence or two to state that an iterative fit was used. We prefer to keep detail where possible in an instrumental paper of this nature.

Line 410: Is the 0.2 pptv 1-sigma? That is correct. We now write: The instrumental noise on the $NO_3$ signal was reduced by averaging over $\approx$ 3 s per data-point ($\approx$ 1800 ring-down-events) to give a noise limited detection limit (1 $\sigma$) of ~ 0.2 pptv.

Line 514, Figure 12a: Figure would be much more effective with the y-axis on a log scale to illustrate the difference between night and day. It is difficult to see the nighttime reactivity on this scale, and the quantitative day / night contrast should be of interest to the readers.
We have added a Figure to the supplementary information in log format.